# Structural insights into substrate recognition by the SOCS2 E3 ubiquitin ligase

Wei-Wei Kung[1,2], Sarath Ramachandran[1,2], Nikolai Makukhin[1,2], Elvira Bruno[1] & Alessio Ciulli [1]

The suppressor of cytokine signaling 2 (SOCS2) acts as substrate recognition subunit of a Cullin5 E3 ubiquitin ligase complex. SOCS2 binds to phosphotyrosine-modified epitopes as degrons for ubiquitination and proteasomal degradation, yet the molecular basis of substrate recognition has remained elusive. Here, we report co-crystal structures of SOCS2-ElonginB-ElonginC in complex with phosphorylated peptides from substrates growth hormone receptor (GHR-pY595) and erythropoietin receptor (EpoR-pY426) at 1.98 Å and 2.69 Å, respectively. Both peptides bind in an extended conformation recapitulating the canonical SH2 domain-pY pose, but capture different conformations of the EF loop via specific hydrophobic interactions. The flexible BG loop is fully defined in the electron density, and does not contact the substrate degron directly. Cancer-associated SNPs located around the pY pocket weaken substrate-binding affinity in biophysical assays. Our findings reveal insights into substrate recognition and specificity by SOCS2, and provide a blueprint for small molecule ligand design.

[1] Division of Biological Chemistry and Drug Discovery, School of Life Sciences, University of Dundee, James Black Centre, Dow Street, Dundee DD1 5EH, UK.
[2] These authors contributed equally: Wei-Wei Kung, Sarath Ramachandran, Nikolai Makukhin. Correspondence and requests for materials should be addressed to A.C. (email: a.ciulli@dundee.ac.uk)

Cytokines are small glycoproteins that play important roles in the differentiation, development and function of lymphoid and myeloid cells[1]. The Janus kinase (JAK)—signal transducer and activator of transcription (STAT) signaling pathway plays a critical role enabling cells to respond to specific cytokines by regulating gene expression. Suppressor of cytokine signaling (SOCS) proteins, which comprise of cytokine inducible SH2-containing protein (CIS) and SOCS1–SOCS7, negatively regulate cytokine receptors and inhibit the JAK-STAT signaling pathway[2].

SOCS proteins share a conserved domain architecture comprising of an N-terminal extended SH2 subdomain (ESS) that functions as a bridge at the interface between the SH2 domain and the SOCS box enabling ubiquitination of captured substrate[3–5], followed by a central Src-homology 2 (SH2) domain that recognizes a phosphotyrosine (pY) containing sequence[6], and a C-terminal SOCS box that interacts with the adapter ElonginB-ElonginC complex (EloBC)[7–9]. All SOCS proteins bind to EloBC and recruit Cullin5 with high specificity, forming different SOCS-EloBC-Cullin5-Rbx2 (CRL5$^{SOCS}$) E3 ligases that catalyze ubiquitin transfer and subsequent proteasomal degradation of specific substrates, as a mechanism to regulate diverse biological processes[10–13]. SOCS proteins serve as substrate recognition modules that impart substrate specificity to each CRL5$^{SOCS}$ E3 complex.

Expression of SOCS proteins is induced by cytokine stimulation. Upon cytokine binding, the oligomerized receptors activate the JAK family kinases that phosphorylate specific tyrosine residues on the receptor, including the docking sites for the STAT proteins. The docked STAT proteins are sequentially phosphorylated, they dimerize and translocate into the nucleus, initiating gene transcription of several downstream proteins including the SOCS proteins. SOCS proteins suppress the JAK-STAT pathway via three distinct but often concomitant mechanisms: (1) KIR mediated direct JAK inhibition[3,14]; (2) Blocking STAT activation by competing for receptor pY sites[15]; (3) Targeting the receptor for proteasomal degradation via SOCS E3 ligase activity[16,17]. Some of the SOCS-substrate interactions have been structurally characterized, including SOCS1-JAK[18], SOCS3-gp130[4,19], SOCS3-gp130-JAK2[20], and SOCS6-cKit[21].

SOCS2, one of the members of the SOCS family, is implicated in disorders of the immune system, central nervous system and cancer, and is thus emerging as a promising therapeutic target[22–25]. SOCS2 has been shown as the primary suppressor of growth hormone (GH) pathway where a gigantism phenotype was observed in a SOCS2$^{−/−}$ mice[26]. Paradoxically, the SOCS2 overexpressed transgenic mice also led to the same phenotype[27]. Attenuation of GHR signaling relies on two phosphorylation sites at GHR that are recognized by SOCS2[16,28]. The pY487 site of GHR interacts with CRL5$^{SOCS2}$ E3 ligase that targets the GHR for ubiquitination and proteasomal degradation[16]. A downstream pY595 site interacts with SOCS2, STAT5b and SHP2 (SH2 domain-containing phosphatase 2), enabling SOCS2 to inhibit the signaling by blocking this receptor site from STAT5b[15,27,29,30]. Nonetheless, deletion of both sites is required to remove the inhibitory effect of SOCS2 on the GH signaling[16,28]. Analysis of the binding affinity of SOCS2 for these two phosphorylation sites of GHR reveals that the pY595 region exhibits a higher affinity towards SOCS2 ($K_D$ = 1.6 μM) compared to the pY487 region ($K_D$ = 11.3 μM)[5,31,32]. An 11-mer phosphorylated peptide spanning the pY595 region of GHR was sufficient to pull down the whole CRL5$^{SOCS2}$ complex from human cell lysates[31] as well as CIS, the closest homolog to SOCS2 from the same family, which plays a role in anti-tumor immunity controlling the differentiation of CD4 T helper cell, and the IL-2 and IL-4 response[31,33]. In addition to GHR, other substrates have been identified to interact with SOCS2, including the erythropoietin receptor (EpoR) at pY426[34], the leptin receptor at pY1077[35], the

epidermal growth factor receptor[36] and the insulin-like growth factor-I receptor[37]. The first crystal structure of SOCS2-ElonginB-ElonginC (SBC) was reported in 2006[32], however the structural basis for substrate recognition by SOCS2 has yet remained elusive.

Here, we determine the co-crystal structures of SBC in complex with phosphorylated epitope peptides from its physiological targets GHR and EpoR. Our structures reveal the peptides are accommodated in an extended conformation to capture specific interactions with SOCS2. A key flexible region of SOCS2, known as the BG loop, is defined in the electron density and shown not to contact the bound substrates. Structural analyses supported by biophysical and mutagenesis investigations identify hotspot residues on the substrate degrons and functionally elucidate disease-relevant single nucleotide polymorphisms (SNPs) of SOCS2. Our findings reveal fresh insights into the molecular recognition and selectivity between SOCS2 and target substrates, and provide an important template for future structure-guided ligand design.

## Results

**Crystallization of substrate-bound SOCS2.** To elucidate the molecular basis of substrate recognition by SOCS2, we subjected the SOCS2-ElonginB-ElonginC (SBC) complex to extensive co-crystallization trials with 11-residue phosphopeptides of either EpoR or GHR that span the regions surrounding Tyr426 and Tyr 595 region, respectively. The affinity of SBC for EpoR_pY426 ($K_D$ of 6.9 μM) and GHR_pY595 ($K_D$ of 1.1 μM) was measured by isothermal titration calorimetry (ITC), and found to be consistent with the literature[31,32] (Supplementary Fig. 1). Attempts to co-crystallize wild-type SBC protein constructs[31,32] with either GHR or EpoR peptide were unsuccessful as resulting crystals only diffracted poorly. To improve crystal quality, we engineered a cluster of three mutations K115A/K117A/Q118A on SOCS2 that was predicted to significantly reduce surface conformational entropy and thermodynamically favor crystal packing[38]. Crystallization attempts with this new S$^{KKQ}$BC triple-mutant construct (K115A/K117A/Q118A on SOCS2) eventually yielded high-resolution datasets.

**SBC-EpoR co-crystal structure.** The structure of SBC in complex with EpoR_pY426 peptide (SBC-EpoR) was solved and refined at 2.69 Å with 19.64% $R_{work}$ and 23.51% $R_{free}$ (Table 1). The overall subunit and domain arrangements of the SBC-EpoR structure is consistent with those of the apo SBC structures[32,39,40] (Fig. 1a). Electron density for nine out of eleven non-terminal EpoR_pY426 residues are well defined in the structure (Fig. 1b). A classic SH2 domain-pY peptide interaction is observed, where the pY residue is anchored at the pY pocket and the flanking residues are extending across the SH2 domain. The pY residue is tightly locked by an intricate hydrogen-bonding network formed by residues Arg73, Ser75, Ser76, Thr83, and Arg96 of SOCS2 (Fig. 1c). Additional hydrogen bonds are formed along the backbone of EpoR_pY426 peptide from Glu(−1) to Leu(+3) with SOCS2 residues Thr93, Asn94, Asp107 and one structural water (Fig. 1d). Multiple hydrophobic interactions also support the binding of EpoR_pY426 C-terminal residues, Ile(+2), Leu(+3), and Pro(+5) that are well accommodated within a hydrophobic patch created by Leu95, Leu106, Ser108, Ile109, Val112, Leu116, and Leu150 of SOCS2 (Fig. 1e).

**Crystal structure of SBC-GHR.** Encouraged by the success in solving an SBC-EpoR structure, to deepen understanding of the SOCS2 binding epitopes, we co-crystallized SBC with an 11-mer GHR_pY595 peptide (SBC-GHR) and solved the structure at 1.98 Å resolution with 19.00% $R_{work}$ and 22.66% $R_{free}$ (Table 1). In contrast to SBC-EpoR, which contains one protomer in the

**Table 1 Crystallographic data collection and refinement statistics**

|  | SBC-EpoR | SBC-GHR | SBC-GHR$_2$ |
|---|---|---|---|
| PDB code | 6I4X | 6I5N | 6I5J |
| Data collection |  |  |  |
| Wavelength (Å) | 0.9686 | 0.9795 | 0.9686 |
| Space group | I 1 2 1 | P 2$_1$ 2$_1$ 2 | P 2 2$_1$ 2$_1$ |
| Cell dimensions |  |  |  |
| $a, b, c$ (Å) | 41.29, 56.33, 203.39 | 113.18, 156.76, 57.57 | 57.83, 113.71, 156.94 |
| $\alpha, \beta, \gamma$ (°) | 90.00, 91.53, 90.00 | 90.00, 90.00, 90.00 | 90.00, 90.00, 90.00 |
| Molecules/ASU | 1 | 2 | 2 |
| Resolution | 29.36–2.69 (2.82–2.69) | 113.55–1.98 (2.01–1.98) | 92.08–2.80 (2.95–2.80) |
| $R_{merge}$ (%) | 10.8 (51.4) | 9.4 (103.5) | 19 (72.1) |
| <I/σ (I)> | 9.6 (2.4) | 17.1 (2.2) | 7.3 (2.7) |
| Completeness (%) | 93.2 (63.3) | 100 (100) | 100 (100) |
| Redundancy | 4.9 (4.2) | 13.3 (13.2) | 7.8 (7.8) |
| CC$_{1/2}$ | 0.99 (0.82) | 1.0 (0.9) | 0.98 (0.74) |
| Refinement |  |  |  |
| Resolution (Å) | 2.69 | 1.98 | 2.8 |
| Unique reflections | 12,265 (791) | 72,170 (7105) | 26,275 (2606) |
| $R_{work}/R_{free}$ (%) | 19.64/23.51 | 19.00/22.66 | 20.96/26.38 |
| Wilson B factor (Å$^2$) | 44.1 | 24.2 | 42.5 |
| Average B factor (Å$^2$) | 46.9 | 32.0 | 42.5 |
| No. non-hydrogen atoms Protein/ligand/water | 2688/83/11 | 6433/344/512 | 5775/327/33 |
| R.M.S.D. |  |  |  |
| Bond lengths (Å) | 0.002 | 0.010 | 0.002 |
| Bond angles (°) | 0.451 | 1.064 | 0.412 |
| Ramachandran analysis |  |  |  |
| Preferred regions (%) | 96.12 | 97.82 | 96.83 |
| Allowed regions (%) | 3.58 | 2.18 | 3.17 |
| Outliers (%) | 0.3 | 0.00 | 0.00 |

Values in parentheses are for the highest resolution shell

asymmetric unit, the SBC-GHR contains two copies of protomer. Alignment of these two protomers via the backbone atoms of the EloB subunit reveals a hinge motion between the SH2 domain and the SOCS box (Supplementary Fig. 2). Such motion is a common feature in SOCS box and F-box containing proteins and it has been shown to be important as it facilitates accurate orientation and positioning of a target substrate protein relative to the multisubunit CRL complex[39,41]. Contrary to the SBC-EpoR complex, which contains a single copy of peptide per SH2 domain, two copies of GHR_pY595 peptides were found binding per SH2 domain of SOCS2 (giving a total of four copies within the asymmetric unit). The two peptides run in an anti-parallel direction relative to each other across the SH2 domain, with well-defined electron density surrounding them both (Fig. 2a). One of the peptides (referred to as peptide A hereafter) binds to SH2 domain in a canonical manner, where the pY is recognized by the positively charged pY pocket between the central β strands and αA (Fig. 2b). In contrast, the second peptide, peptide B, has its pY residue exposed to solvent and interacting only with His149 of SOCS2 (Fig. 2b).

**Specific interaction of the GHR_pY595 phosphopeptide.** The unusual simultaneous binding observed for the GHR substrate peptide to SOCS2 SH2 domain is imparted mainly by the region comprising Ser(+2) to Val(+6) from each peptide, which pair such that they form an anti-parallel beta sheet (Fig. 2c). Extensive hydrogen bonds are formed between the backbone of the two peptides and backbone residues of SOCS2 and structural waters (Fig. 2c). Further hydrophobic interactions appear to reinforce the binding, which impart specificity for GHR. The Ile(+3) and Ile(+5) of peptide A and peptide B settle in a hydrophobic patch of the SH2 domain formed by

Leu95, Leu106, Ser108, Leu116, and Leu150 (Fig. 2d). Another hydrophobic interaction that is distinct in SBC-GHR compared to SBC-EpoR is that formed by the side chain of Val(−3) of peptide A, that nicely fits into a hydrophobic pocket comprising of Thr88, Ala90, Thr93, Leu95, and Val148 from SOCS2 (Fig. 2e). In the SBC-GHR structure, a cobalt ion is modeled at a positive peak that disappeared only at 21 σ level in the unbiased Fo-Fc electron density map. This cobalt ion satisfies the formation of an octahedral coordination geometry with the side chains of His(+4) of peptide B, His149 of SOCS2 and with four surrounding water molecules (Fig. 2f).

**The BG loop of SOCS2 is observed in an open conformation.** SOCS2 recognizes two GHR binding sites at regions around pY487 and pY595, respectively. We therefore hypothesized that the two copies of the GHR peptides bound in the crystal structure might mimic a physiological folded conformation of GHR, presenting each of the phosphorylated sites bound simultaneously to SOCS2. To test this hypothesis, we utilized an 11-residue GHR_pY487 phosphopeptide (NIDFpYAQVSDI, K$_D$ of 2.3 μM by ITC, Supplementary Fig. 1), mixed with the GHR_pY595 peptide and SBC in equimolar 1:1:1 ratio for co-crystallization. In this crystal structure (hereafter referred to as SBC-GHR$_2$), still two copies of the GHR_pY595 peptide, but no GHR_pY487, are observed bound, yielding a structure very similar to the previous SBC-GHR structure (Supplementary Fig. 3). However, an important observation in SBC-GHR$_2$ compared to our other co-crystal structures was that the region of SOCS2 corresponding to residues 134–162, also called BG loop, is now fully visible in the electron density. The BG loop connects the αB and βG strand of an SH2 domain (Fig. 3a). The first part of the BG loop (residues 134–148 in SOCS2) differs in length and sequence among SOCS

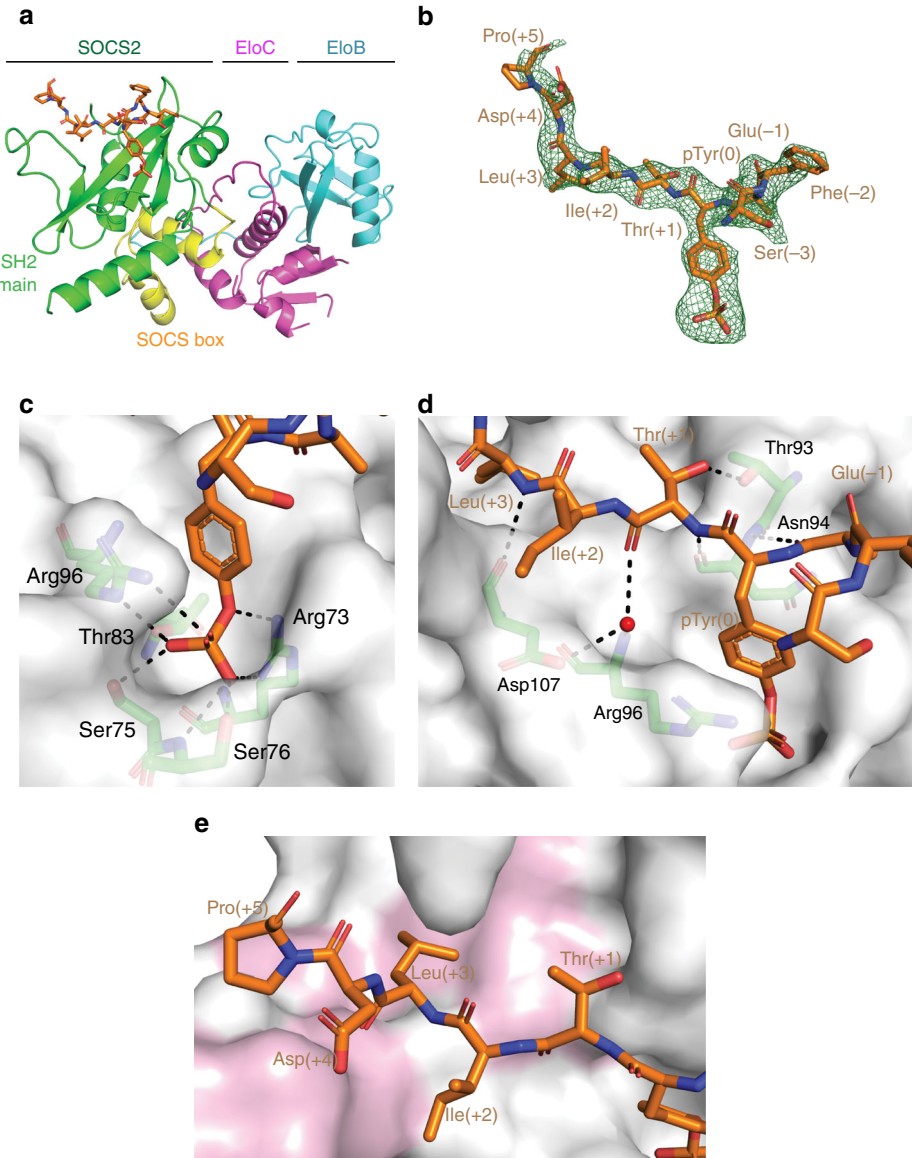

**Fig. 1** Structural and interaction detail of the SBC-EpoR co-crystal. **a** Domain and subunit arrangement of the SBC-EpoR co-crystal. Protein chains are shown in cartoon, with EloB (cyan), EloC (magenta) and SOCS2, comprising of SOCS box (yellow) and SH2 domain (green). The EpoR_pY426 peptide is shown in orange stick. **b** The Fo-Fc ligand omit map of the EpoR_pY426 peptide (green mesh) contoured at 2.0 σ level to highlight densities for the EpoR_pY426 peptide (orange stick). **c** Hydrogen bond interactions (dash) between the pY of EpoR_pY426 peptide (orange stick) and SOCS2 residues (green stick). **d** Hydrogen bond interaction (dash) between the EpoR_pY426 peptide (orange stick), SOCS2 (green stick) and water (red sphere). **e** Hydrophobic interaction between EpoR_pY426 peptide (orange stick) and SOCS2 (surface). Hydrophobic residues on SOCS2 are colored in pink

proteins (Fig. 3a, b). We refer herein to this more variable region as the "specificity BG loop", because its conformation, together with that of the adjacent EF loop, governs accessibility of the pY binding pocket and contributes to substrate specificity in SH2 domains[42] (Fig. 3b). A particular region in the middle of the specificity BG loop (residues 136–145) is found to be disordered in all previously determined SOCS2 structures (PDB code, 2C9W [https://doi.org/10.2210/pdb2C9W/pdb], 4JGH [https://doi.org/10.2210/pdb4JGH/pdb] and 5BO4 [https://doi.org/10.2210/pdb5BO4/pdb]) as well as our other co-crystal structures SBC-GHR and SBC-EpoR. In this SBC-GHR₂ structure, the BG loop is in an open conformation stabilized by crystal contacts, Pro140 (BG loop) to Arg186 (SOCS box) and Pro140 (BG loop) to Ile90 (EloB), as clearly defined by the unbiased omit map at this region (Fig. 3c; Table 1).

**Conformational changes of the EF and BG loop**. The configurations of the EF and BG loops play an important role in governing the accessibility of the binding pocket and specificity toward ligand binding[42]. A comparison of SOCS2 structures in the presence and absence of peptides bound highlight conformational changes in EF (residue 107–116) and BG loop. In the absence of substrate peptide, the EF loop curls up placing the Ile110 and Cys111 at the hydrophobic SH2 domain (Fig. 4a). Upon binding of a substrate peptide, the EF loop opens up forming backbone interactions with GHR_pY595 (Fig. 4b), or rearranges itself to allow a specific interaction with EpoR_pY426 (Fig. 4c). This specific interaction between EF loop and EpoR_pY426 involves hydrophobic interactions between Ile109 and Val112 of SOCS2, Val112 of a SOCS2 symmetry mate, and Pro(+5) of EpoR_pY426, resulting in a differential binding mode

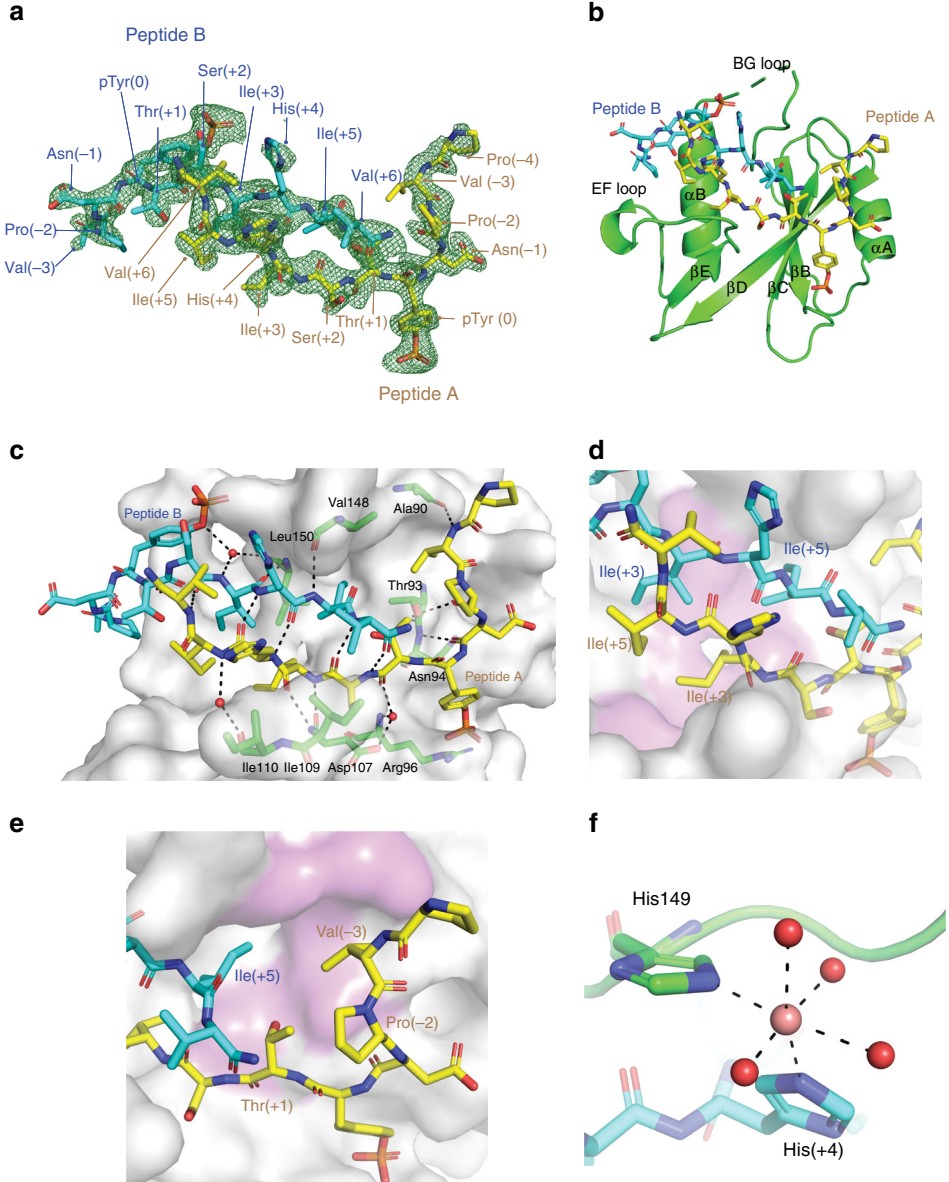

**Fig. 2** Structural and interaction detail of the SBC-GHR co-crystal structure. Two copies of GHR_pY595 were observed in the co-crystal. One copy is shown as peptide A (yellow stick) and the other one as peptide B (cyan stick). **a** The Fo-Fc ligand omit map of the peptides (green mesh) contoured at 2.5 σ level to highlight densities for the peptide A and peptide B. **b** Cartoon diagram of the SH2 domain of SOCS2 (green) with peptides A and B bound. **c** Hydrogen bond interactions (dash) among peptide A, peptide B, SOCS2 residues (green stick) and water (red sphere). **d**, **e** Hydrophobic interaction of the peptides with C-terminal half and N-terminal half of the SH2 domain (surface), respectively. SOCS2 residues involved in hydrophobic interactions are colored in pink. **f** The coordination of cobalt ion (pink sphere) with His149 of SOCS2 (green), His(+4) of peptide B and water molecules (red sphere)

between EpoR_pY426 and GHR_pY595 to SOCS2. The BG loop of SOCS2 is observed in an open conformation in the SBC-GHR$_2$ structure whilst it is disordered in SBC-EpoR. A superimposition of the two substrate complex structures suggest that the BG loop opens up further in the SBC-GHR structure to accommodate two GHR peptides (Fig. 4d).

**Biophysical characterization of specificity between GHR_pY595 and SOCS2.** To evaluate the specificity of the protein–peptide interaction in solution, we designed single-point mutations on the peptide first, and compared their binding to wild-type peptide by two orthogonal biophysical methods: a direct binding assay using SPR (SBC immobilized on the chip) and $^{19}$F ligand-observed displacement NMR assays. In the $^{19}$F NMR displacement assay, the fluorine signal of a purposely-designed

reporter ligand (also referred to as spy molecule) was monitored as a mean to quantify the extent of the competition between the tested peptides and the spy molecule. A Carr-Purcell-Meiboom-Gill (CPMG) pulse sequence was applied to estimate the spin–spin relaxation time ($T_2$) of the spy molecule in the absence and presence of protein[43–45]. By adding competitor to disrupt the protein-spy interaction, the binding affinity of a competitor can be calculated based on the degree of displacement of the spy[46,47]. The spy molecule used in our assay is compound **3**, a fluorinated pY analog that specifically binds to the pY pocket with a $K_D$ of 50 μM (Supplementary Fig. 4). The two assays were found to be robust and reliable, as the measured $K_D$ (SPR) and $K_i$ (NMR) values correlated well ($R^2$ of 0.74) (Supplementary Fig. 5).

First, we focused on the unique interaction formed by Val(−3) of GHR_pY595, which inserts into a small hydrophobic cavity of

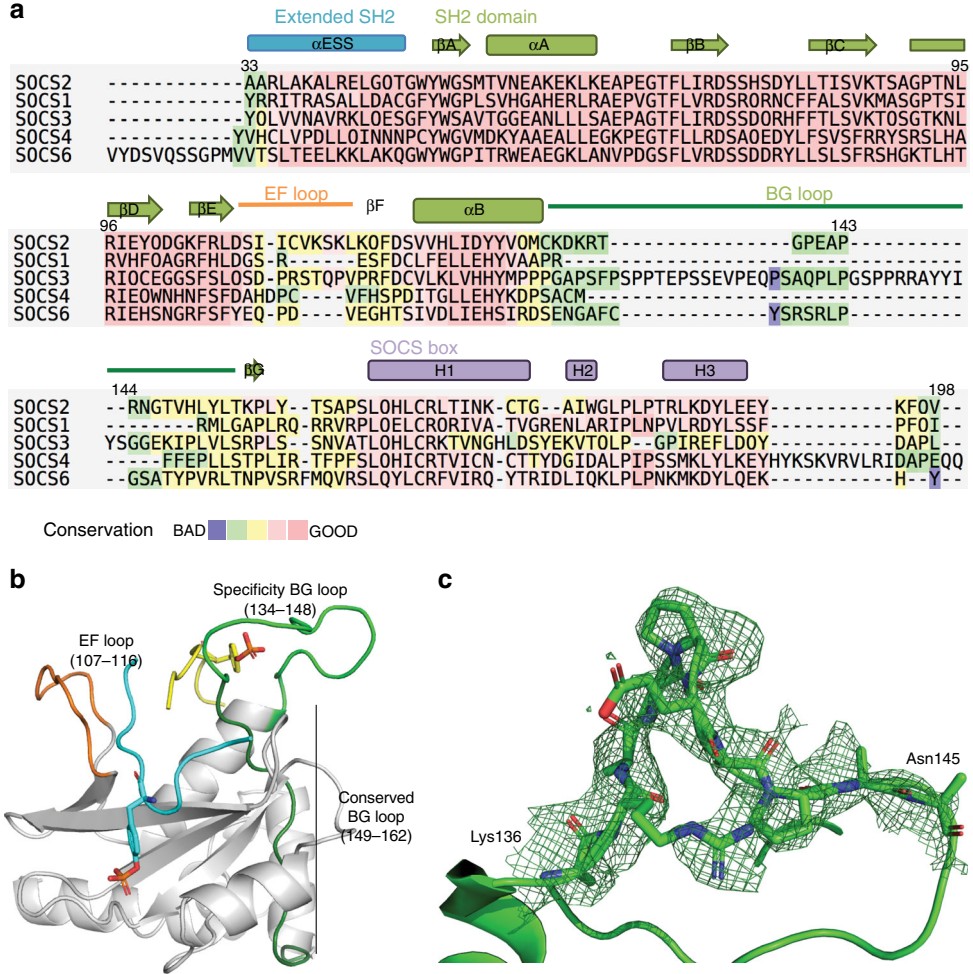

**Fig. 3** The BG loop of SOCS2. **a** Secondary structure elements in SOCS2 are shown above the sequence alignment, numbering of residue is for SOCS2. SOCS proteins with crystal structures available were aligned using T-Coffee expresso mode for sequence alignment with structural information[87]. **b** Locations of the EF loop (orange) and the BG loop (green) on SOCS2 (white) crystal structure with bound GHR peptides A and B (cyan and yellow, respectively). **c** The Fo-Fc omit map of the previously disordered region of the BG loop (green mesh) contoured at 1.5 σ level

SOCS2 (Fig. 2e). This interaction was investigated by mutating Val(−3) in the GHR peptide to Tyr and Arg, as representative bulky and charged residues, respectively. We hypothesized this structural change would disrupt the fit at this small hydrophobic pocket. Mutant V(−3)R exhibited between a 6-fold and a 14-fold loss of binding affinity to SBC, depending on the assay, suggesting the charged group strongly disrupts the interaction (Table 2). By contrast, the V(−3)Y was less disruptive, with only a two-fold loss in affinity.

Next, to map the relative importance and contribution of each individual amino acids to the binding affinity with SBC, alanine scan of the substrate peptides was invoked. The peptide sequences were designed such that individual amino acids were separately mutated into alanine except pY, which is known to abolish binding if mutated even to unphosphorylated Y[31]. The resulting library comprised of the original wild-type sequences, ten derivatives from GHR_pY595 and nine from EpoR_pY426, and was characterized in parallel using SPR and $^{19}$F NMR competition assay (Table 2). Alanine substitution at pY(−3), pY(−1), pY (+3), and pY(+4) of the GHR_pY595 resulted in at least two-fold weakened binding (increase in $K_D$) compared to the wild-type. In contrast, a similar (at least two-fold) weakening in binding affinity was observed in the EpoR_pY426 peptide upon alanine substitution at pY(−1), pY(+2), and pY(+3). These results are

consistent with observations from our crystal structures that peptide-SOCS2 binding is mediated by hydrophobic interaction including pY(−3), pY(+3), and pY(+5) on the GHR_pY595 and pY(+2) and pY(+3) on the EpoR_pY426. The binding affinity for each peptide also dropped by at least two-fold with the Thr/Ala substitution at pY(−1) position, indicating the importance of the residue just upstream of pY.

**SNPs study.** Several single nucleotide polymorphisms (SNPs) on SOCS2 are reported in the Catalog of Somatic Mutation in Cancer database (COSMIC) as potentially linked to cancers such as tumors of the lung, breast, and pancreas[48]. We thus next decided to characterize the interaction of selected SNP SOCS2 mutants with substrate peptides GHR_pY595, EpoR_pY426 and GHR_pY487 by SPR (Table 3). Inspection of our SBC co-crystal structures guided us to select five known SNPs: N94D, R96L, and R96Q that are located in the pY-pocket and involved in direct recognition of pY; L106V that is located in the hydrophobic patch of the SOCS2 SH2 domain that is involved in substrate interaction; and C133Y that participates in the SH2 hydrophobic core (Fig. 5a). All mutant proteins expressed and purified similarly to wild-type, and the mutations did not appear to affect the structural integrity and solubility of the constructs, as observed by $^1$H NMR (Supplementary Fig. 6).

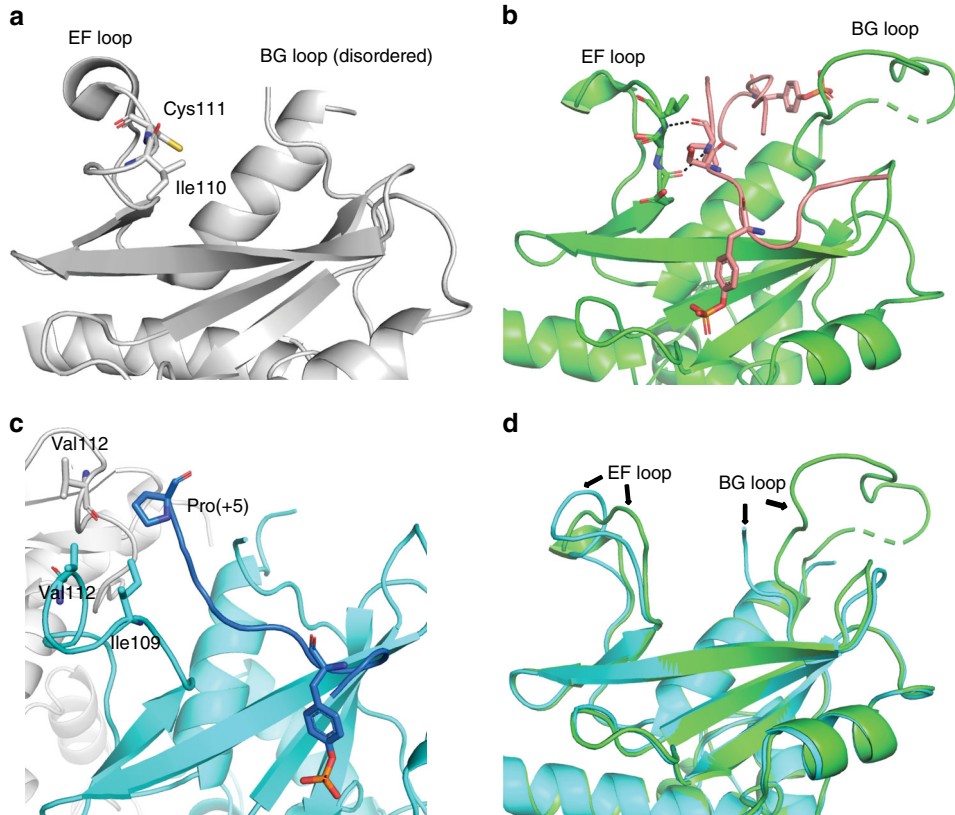

**Fig. 4** Conformational changes of the EF and BG loops. **a** The EF loop curls up in the apo SOCS2 structure (white, PDB code: 2C9W), placing Ile110 and Cys111 (stick) at the SH2 hydrophobic pocket. **b** The EF loop makes backbone interactions with GHR peptide (pink) in the SBC-GHR₂ structure (green). **c** The Ile109 and Val 112 of SOCS2 (cyan) and Val112 of SOCS2 symmetry mate (white) make unique hydrophobic interaction with Pro(+5) of EpoR peptide (blue). **d** Superposition of SOCS2 from SBC-GHR₂ (green) and SBC-EpoR (cyan) displays conformational changes of the BG and EF loops

---

**Table 2 Affinity measurements for GHR_pY595 and EpoR_pY426 wild-type and mutant peptides binding to SBC**

| Sequence | Peptide | SPR $K_D$ (μM) | NMR $K_i$ (μM) |
|---|---|---|---|
| PVPDpYTSIHIV | GHR wild-type | 1.5 ± 0.09 | 1.5 ± 0.06 |
| PRPDpYTSIHIV | V(−3)R | 9.0 ± 1.05 | 20.6 ± 2.09 |
| PYPDpYTSIHIV | V(−3)Y | 3.3 ± 0.13 | 3.3 ± 0.24 |
| AVPDpYTSIHIV | pY(−4) | 1.7 ± 0.14 | 0.9 ± 0.17 |
| PAPDpYTSIHIV | pY(−3) | 2.7 ± 0.19 | 3.4 ± 0.25 |
| PVADpYTSIHIV | pY(−2) | 1.4 ± 0.12 | 4.1 ± 0.75 |
| PVPApYTSIHIV | pY(−1) | 6.2 ± 0.37 | 21.7 ± 3.34 |
| PVPDpYASIHIV | pY(+1) | 1.0 ± 0.09 | 0.5 ± 0.07 |
| PVPDpYTAIHIV | pY(+2) | 1.4 ± 0.07 | 1.2 ± 0.06 |
| PVPDpYTSAHIV | pY(+3) | 3.7 ± 0.22 | 3.4 ± 0.66 |
| PVPDpYTSIAIV | pY(+4) | 6.7 ± 0.78 | 18.5 ± 2.67 |
| PVPDpYTSIHAV | pY(+5) | 2.4 ± 0.13 | 1.4 ± 0.36 |
| PVPDpYTSIHIA | pY(+6) | 1.9 ± 0.09 | 1.6 ± 0.21 |
| ASFEpYTILDPS | EpoR wild-type | 13.3 ± 0.59 | 7.1 ± 1.01 |
| AAFEpYTILDPS | pY(−3) | 14.2 ± 0.65 | 7.5 ± 1.04 |
| ASAEpYTILDPS | pY(−2) | 16.3 ± 1.02 | 7.2 ± 0.42 |
| ASFApYTILDPS | pY(−1) | 21.7 ± 1.49 | 23.5 ± 2.99 |
| ASFEpYAILDPS | pY(+1) | 17.1 ± 0.38 | 10.3 ± 1.28 |
| ASFEpYTALDPS | pY(+2) | 30.8 ± 1.24 | 17.2 ± 0.89 |
| ASFEpYTIADPS | pY(+3) | 32.3 ± 1.42 | 20.4 ± 0.88 |
| ASFEpYTILAPS | pY(+4) | 18.0 ± 0.91 | 14.1 ± 1.74 |
| ASFEpYTILDAS | pY(+5) | 8.9 ± 1.25 | 6.9 ± 1.39 |
| ASFEpYTILDPA | pY(+6) | 16.4 ± 0.29 | 11.3 ± 1.29 |

Values reported are the means ± s.e.m. from four independent experiments

---

**Table 3 SPR dissociation constants $K_D$ (μM) of substrate peptides for SNP mutants of SOCS2**

| Protein | GHR_pY595 | GHR_pY487 | EpoR_pY426 |
|---|---|---|---|
| Wild-type | 1.4 ± 0.12 | 5.1 ± 0.77 | 12.5 ± 0.63 |
| L106V | 2.4 ± 0.11 | 8.9 ± 0.70 | 14.1 ± 1.23 |
| C133Y | 2.2 ± 0.10 | 7.7 ± 0.71 | 13.4 ± 0.82 |
| N94D | Weak | Weak | n.d. |
| R96L | Weak | Weak | n.d. |
| R96Q | n.d. | n.d. | n.d. |

Values reported are the means ± s.e.m. from four independent experiments
Weak: signal detected but saturation was not achieved
n.d.: signal not detected

The L106V and C133Y mutations did not affect binding affinities compared to wild-type (Table 3). In contrast, the N94D and R96L mutations drastically impaired substrate binding, leading to almost undetectable binding response by SPR. Because of the low signal-to-noise, reliable $K_D$ values could not be measured with these protein mutants. For the R96Q mutation no signal response was detected, suggesting that binding was completely abolished and highlighting the most disruptive of the mutations studied herein. Our result is consistent with evidence by Rupp et al. that the point mutation R96C abrogates substrate binding by SOCS2[49].

In addition to SPR, we evaluated to what extent SNP mutants retain competence to bind substrate pY by monitoring the transverse relaxation rate ($R_2$) of spy molecule **3** using ¹⁹F NMR

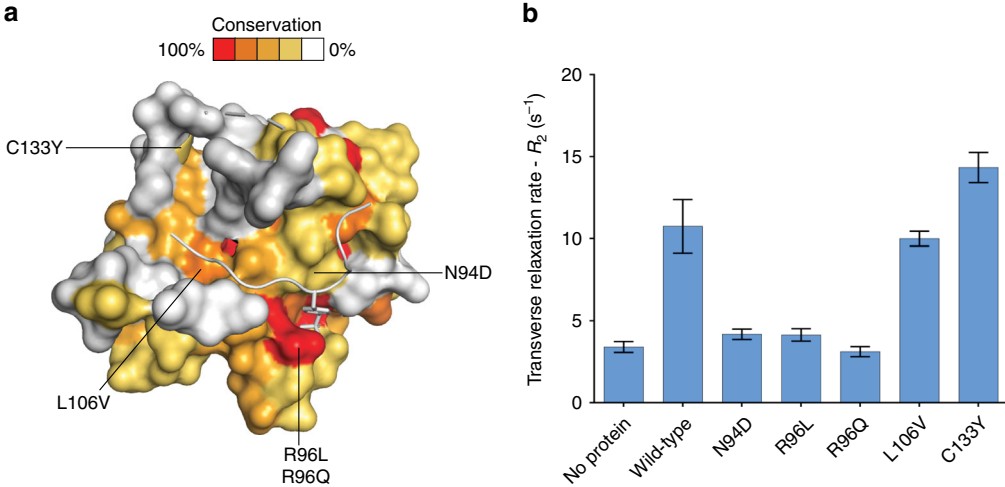

**Fig. 5** Position and characterization of SNPs of SOCS2. **a** Sequence conservation mapped onto the SH2 domain of SOCS2 (shown as surface). Conservation surface representation based upon the ClustalW multiple sequence alignment of CIS and SOCS1–SOCS7 sequences where highly conserved residues are shown in red/orange color and variable residue positions colored white/gold. Positions of SNP mutations are highlighted. **b** Transverse relaxation rates ($R_2$) of spy molecule **3** binding to SBC protein variants. The $R_2$ relaxation rates were obtained by fitting as exponential decay the $^{19}F$ peak integrals of **3** measured at five CPMG delays in a sample in the absence or presence of SBC protein (see Supplementary Fig. 7). Error bars reflect the quality of the fit between the non-linear least-square curve and the experimental data

spectroscopy (Fig. 5b and Supplementary Fig. 7). Spy molecule **3** exhibited $R_2$ values of $3 \, s^{-1}$ when free in solution, while its $R_2$ increased to $11 \, s^{-1}$ in the presence of wild-type SBC, providing a good assay window. The $R_2$ values were comparable to wild-type in the presence of L106V and C133Y SBC mutants ($10 \, s^{-1}$ and $14 \, s^{-1}$, respectively), indicating that these mutations retain pY recognition. In contrast, mutations on N94 and R96 led to $R_2$ values for spy molecule **3** comparable to those in the absence of protein (around $3–4 \, s^{-1}$), consistent with a loss of pY binding. Together, the $^{19}F$ NMR and the SPR data consistently elucidate abrogated substrate binding for SOCS2 SNP mutations N94D, R96L, R96Q, which are located around the pY pocket.

## Discussion

SOCS2 is the substrate recruiting subunit of a CRL5 E3 complex that negatively regulates the JAK-STAT signaling by targeting substrate receptors for degradation and blocking STAT5b activation by competing with receptor pY sites. The details of these interactions have remained elusive and to date structural information remained limited to apo SOCS2. Herein, we have disclosed structures of SBC in complex with substrate peptides EpoR_pY426 and GHR_pY595. Both peptides recapitulate a canonical substrate-binding mode to the SH2 domain of SOCS2. Residues at pY(−1), pY(+1) and pY (+3) positions of the GHR_pY595 and EpoR_pY426 peptide contains similar properties; whereas residues at pY(−3), pY(+2), and pY(+5) position are different in properties and sizes (Fig. 6a). In particular, the Val(−3) of GHR_pY595 and Pro(+5) of EpoR_pY426 peptide catch different hydrophobic interactions resulting in exclusive binding modes in SOCS2 compared to the substrate peptides bound to SOCS3 and SOCS6 (Fig. 6b, c)[4,21].

The BG loop of SOCS2 had not been fully revealed in previous published structures. Here, we report an open conformation of the BG loop, which differs to other SOCS structures with bound peptides, for example SOCS3:gp130 and SOCS6:c-kit[19,21]. In the SOCS3 and SOCS6 peptide-bound structures, the BG loop folds up as a hairpin interacting with the substrate peptide, forming a triple-stranded β sheet structure (Supplementary Fig. 8a, b). The corresponding BG loop region in SOCS2 is either fully disordered

or in an open conformation (SBC-EpoR and SBC-GHR₂), suggesting that this region does not participate in substrate recognition. Nevertheless, interestingly, a similar triple-stranded β sheet structure is observed in the SBC-GHR₂ structure, where the peptide B replaces the first β-sheet of the BG loop and makes backbone interactions with the BG loop and peptide A (Supplementary Fig. 8c). A tyrosine phosphatase SHP-2, which contains SH2 domains, features a comparable structure[50]. Its BG loop folds up as a hairpin and forms a triple-stranded β sheet interaction with two bound peptides (Supplementary Fig. 8d). Although the canonical phosphotyrosine binding site is conserved for both SOCS2 and SHP2, the positioning of the non-canonical phosphotyrosine varies significantly. The distance between Cα atoms of phosphotyrosines within the bound peptides are 23 Å and 13 Å in SBC-GHR and SHP2-substrate co-structures, respectively (Supplementary Fig. 8e). Longer distance between the canonical and non-canonical pY enable antiparallel beta strand interactions between four amino acids of GHR peptides unlike antiparallel interactions between two amino acids in SHP-2 substrate peptides. Furthermore, a longer EF loop in SOCS2 pushes the two GHR peptide away from itself towards the open BG loop, enabling β sheet formation.

The BG loop along with the EF loop forms a hydrophobic channel in SOCS3 and SOCS6. This channel imparts specificity and restricts the binding of substrates. In contrast the open conformation of BG loop in SOCS2 appears to be critical in enabling SOCS2 to accommodate a wider range of substrates including GHR, EpoR, SOCS1, and SOCS3 amongst others. A comparison of buried surface area of the substrate peptides among SOCS proteins, reveal that EpoR and GHR bind with SOCS2 with only 595 Å² and 641 Å², respectively, in contrast to areas of 1714 Å² for SOCS6/c-KIT and 1761 Å² for SOCS3/gp130 complexes. Unlike SOCS3 and SOCS6 complex structures, the pY flanking residues from EpoR and GHR do not participate in extensive side-chain hydrogen bonding interactions. Together, these observations are consistent with greater binding affinities of SOCS3 and SOCS6 substrates compared to SOCS2 substrates. The lower binding affinity for SOCS2 substrates could contribute to its relatively greater promiscuity to multiple substrates.

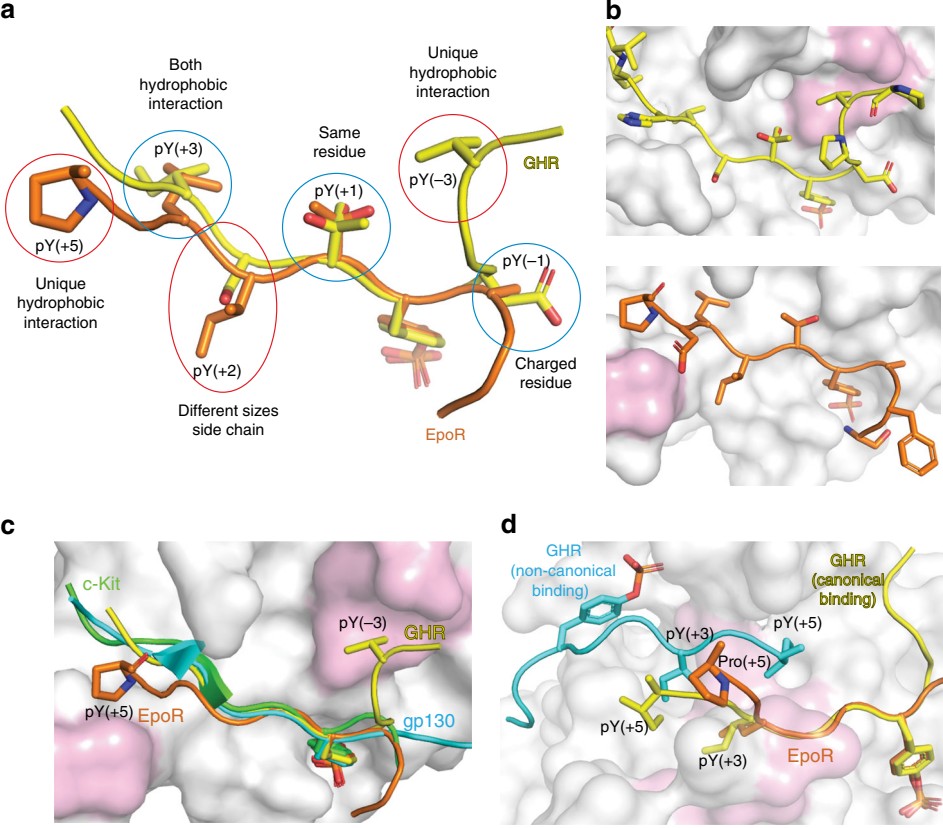

**Fig. 6** Comparison of the GHR and EpoR binding. **a** Overlay of the GHR_pY595 (yellow) and EpoR (orange) peptides as bound to SOCS2. Side chains circled in blue have similar properties and red have different properties. **b** Different hydrophobic interactions (pink) caught by GHR_pY595 (yellow) and EpoR (orange) peptide. **c** Overlay of the substrate peptide of SOCS proteins. The GHR_pY595 (yellow) and EpoR (orange) peptides catch distinct hydrophobic cavities (pink) on SOCS2. These interactions differentiate SOCS2 substrate binding mode from other substrates of SOCS proteins. The SOCS3 substrate peptide gp130 is in cyan (PDB ID: 2HMH) and SOCS6 substrate peptide c-Kit is in green (PDB ID: 2VIF). **d** The binding of the non-canonical GHR (cyan) relies on backbone to backbone interaction and hydrophobic interaction from pY(+3) and pY(+5) of the GHR_pY595 to the SH2 domain hydrophobic core (pink). The binding mode of EpoR (orange) makes it unfavorable to establish these interactions

The observation of the dual-peptide binding mode to SOCS2 was unexpected, however is not unprecedented with SH2 domains, as reported previously with the tyrosine phosphatase SHP-2[50]. In the co-crystal structure of SHP-2:pY peptide solved by Zhang et al., one pY of the peptide is recognized at the pY pocket and the other one is solvent exposed as in our structures described herein[50]. Besides, two peptides run antiparallel to each other and form an antiparallel four-stranded β sheet with the BG loop (Supplementary Fig. 8d). Zhang et al. suggested that the dimerization of peptide binding in SHP-2 requires at least one pY containing peptide and leads to enhanced binding affinity. In the case of the SBC-GHR complex, despite preparing several protein–peptide samples for co-crystallization at 1:1 molar ratio, all dataset collected from crystals were consistent with a 1:2 (protein–peptide) binding mode. In both cases, the binding of the non-canonical binding peptide relies on hydrophobic interaction with the SH2 domain hydrophobic core, and backbone to backbone interaction with the canonical binding peptide and protein (Fig. 6d). The 1:1 binding mode for EpoR could be justified by the presence of Pro(+5) in EpoR, that acts as a strand breaker and prevents backbone to backbone interaction with the second peptide. Indeed, from the alanine scan (Table 2), we observe that when Pro(+5) is replaced by alanine, the affinity improves— implying the possibility of secondary peptide binding.

We put forth two distinct models that might explain the dual peptide recognition mode and its role in specific tuning of GHR signaling response. First, a "cis" recognition mode, where the GHR tail folds back as a hairpin structure presenting two binding epitopes around distinct phosphorylation sites (e.g., pY487 and pY595) for recognition (Fig. 7a). However, the crystallography data from our follow-up experiment as described in the SBC-GHR₂ structure is not consistent with this hypothesis, as two instances of the pY595 peptide were found bound despite a molar ratio of 1:1 for pY487 and pY595 peptides being present in the co-crystallization buffer. However, we cannot exclude that simultaneous recognition of the two distinct epitopes would require a loop of the same tail twisting back onto itself to enhance the binding affinity of second epitope. Alternatively, we envisage a "trans" recognition mode, where SOCS2 recognizes two separate receptor tails of the activated dimerized GHR receptors at the cell membrane (Fig. 7b). SOCS2 might additionally play a role as scaffold bringing two substrates in close proximity, for example by recruiting one instance of phosphorylated substrate to assist the binding of un-phosphorylated one for post-translational modification. This mechanism evokes potential similarities with some phosphodegrons which require two sites to be phosphorylated, utilizing a first kinase to "prime" phosphorylation events, followed by a second kinase for follow-on phosphorylation[51]. An example of such a mechanism is the β-catenin degradation mediated by the β-TrCP[52]. Further biophysical investigation is warranted to address the extent to which these potential mechanisms might be invoked for SOCS2 function.

SOCS2 is an attractive therapeutic target due to its links to cancer, diabetes, neurological and inflammatory diseases[23,24,53–57].

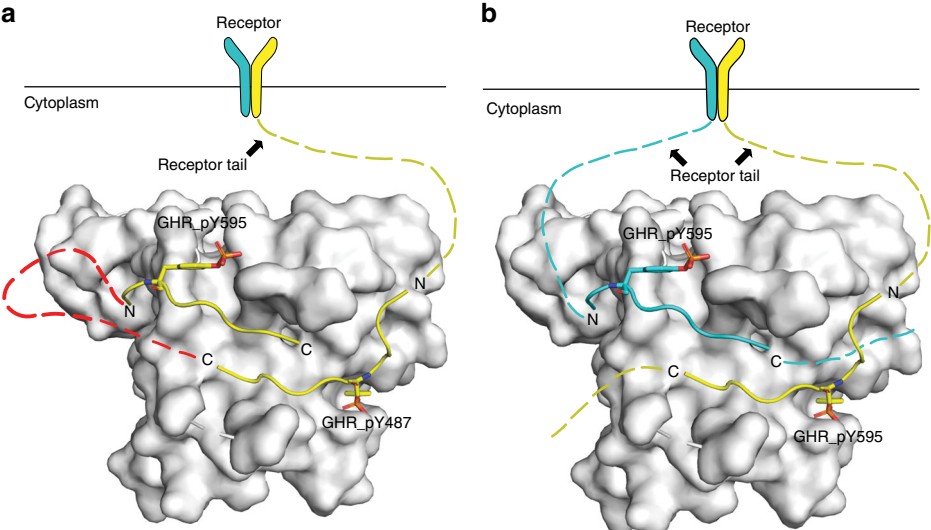

**Fig. 7** Two models of SOCS2-substrate receptor recruitment. **a** Illustration of a "cis" recognition where the two binding sites on the same molecule of GHR, pY487 and pY595, fold into a hairpin structure for SOCS2 binding. SOCS2 is shown in white surface, the hairpin structure is illustrated by connecting the two peptides (yellow cartoon) with a red dashed line. **b** Illustration of a "trans" recognition where SOCS2 recruits two molecules of receptor tails (cyan and yellow dashed lines) simultaneously after dimerization of the receptor at the membrane

Breast, lung, liver, and ovarian cancer have been correlated with downregulation in SOCS2[58–63]. In addition to the JAK-STAT pathway, a recent study has identified the involvement of SOCS2 in the NF-κB (nuclear factor kappa-light-chain-enhancer of activated B cells) pathway that regulates the immune and inflammatory responses[64,65]. NF-κB is found to be constitutively activated in many types of cancer and influences a diverse array of pro-tumorigenic functions, therefore NK-κB plays a pivotal role in cancer initiation and progression[66]. SOCS2 negatively regulates TNFα induced NF-κB activation by targeting NDR1, a serine-threonine kinase, for proteasomal degradation. Hence SOCS2 deficiency may lead to increased levels of NDR1, which results in aggressive behavior of PC3 prostate cancer cells[64]. These evidences highlight the potential in targeting SOCS2 for drug discovery for inflammation and cancer biology. We have revealed structural insights into the SOCS2-peptide interactions by X-ray crystallography and identified hotspot using alanine scanning, mutation study and SNPs study. This information provides a template to guide the structure-based rational design of SOCS2 ligands that are instrumental in the development of novel chemical tools to probe SOCS2 biology, and in the quest for novel small molecule ligands binding to SOCS2 as potential therapeutics. SOCS2 binders at the pY binding pocket can be used as inhibitors of the CRL5$^{SOCS2}$, which would be expected to prevent degradation of target substrate receptors, thus prolonging the activity of cytokine signaling pathway and upregulating expression of endogenous STAT5b-responsive gene expression. In a distinct application, a SOCS2 binder could provide a novel E3 ligase ligand for designing new chemical degraders to hijack SOCS2 CRL activity and trigger the degradation of unwanted proteins inside cell[67–69]. This approach, also known as proteolysis targeting chimeras (PROTACs), offers the advantage of inducing rapid and selective intracellular depletion of the target protein, as opposed to mere blockade of a single inter-action or activity, which pairs more closely to genetic target validation and often results in greater maximal efficacy of intervention in a signaling pathway. PROTAC-mediated protein degradation has been shown to occur at very low compound concentration (pM to nM range), well below the range of inhibitory concentrations for the hijacked E3 ligase[70,71]. PROTACs potentially allows targeting of intractable protein targets that are beyond the reach of conventional small-molecule approaches that require full occupancy of a target

binding site e.g., receptor antagonists and enzyme inhibitors. A limited set of E3 ligases have been targeted so far for PROTACs, notably VHL[70,71] and cereblon[72,73], so extending the approach to other ligandable E3 ligases would be an important advance to the field. The usage of SOCS2 binders for PROTACs may be specific to SOCS2 expressing cells, thus could provide an additional layer of tissue specific degradation of target proteins. For exam-ple, PROTACs designed using SOCS2 binders as recruiters would be ideal for degrading targets specifically in cells affected with leukemia and gastrointestinal sarcoma which have been reported to have upregulated SOCS2 expression[74]. Low levels of SOCS2 in other normal tissues would help minimize toxicity. On the other hand, a pan-SOCS recruiting PROTACs would help diversifying the target tissue range. Our peptide-bound co-crystal structures suggest that SOCS2 might be ligandable and provide a blueprint for the rational structure-guided design of novel SOCS2 inhibitors and SOCS2 ligands for PROTACs.

## Methods

**Cloning and protein expression.** The human SOCS2 (amino acids 32–198) and the ElonginB (amino acids 1–104) and ElonginC (amino acids 17–112) plas-mids were used for protein expression as previously reported[31,39]. Briefly, SBC was co-expressed in *E. coli* BL21(DE3) from pLIC (His₆-SOCS2) and pCDF (EloBC) plasmids. Protein expression was induced with isopropyl β-d-1-thiogalactopyranoside at 18 °C for 12 h. After cell lysis, SBC protein was found in the soluble fraction and purified by affinity chromatography using a HisTrap column (GE Healthcare). Following tag cleavage with tobacco etch virus (TEV) protease and a second HisTrap column the desired untagged protein eluted in the flow-through fractions. SBC was finally purified by size-exclusion chromatography on a Superdex 75 16/600 column (GE Healthcare) in 25 mM HEPES, pH 7.5, 250 mM NaCl and 10 mM DTT. SOCS2 mutants N94D, R96L, R96Q, L106V or C133Y were introduced using PCR-based site-directed mutagenesis (details of the primers used are in Supplementary Table 1). SBC containing mutant SOCS2 were co-expressed and purified as described above.

**Crystallization and structure determination of SBC-GHR.** To improve crystal-lization, surface entropy reducing mutations were introduced into SOCS2 construct (amino acids 32–198). Three mutation clusters (K63A/E64A/E67A; K113A and K115A/K117A/Q118A) were identified with the SER server[38]. SER-assisted crys-tallization attempts yielded crystals with the K115A/K117A/Q118A SOCS2-EloBC (S$^{KKQ}$BC). Five times molar excess of GHR_pY595 (PVPDpYTSIHIV-amide, 5 mg ml⁻¹) was incubated with S$^{KKQ}$BC, followed by removing unbound peptide using a protein concentrator. Sample was concentrated to 22 mg ml⁻¹ with an additional 0.1 M of sodium cacodylate pH7.2 added to the sample. Diffraction-quality crystals were obtained with 0.005 M Cobalt (ll) chloride, 0.1 M MES pH 6.5,

1.0 M ammonium sulfate at 4 °C using hanging drop vapor diffusion method at 2:1 protein:precipitant ratio. Crystals were cryo-protected using 20% MPD prior to vitrification in liquid nitrogen.

Diffraction data were collected at 100 K at Diamond Light Source beamline i04 using Pilatus 6M-F detector at 0.98 Å wavelength. Indexing and integration was processed by XDS[75] and scaling and merging with AIMLESS within the CCP4 program suite[76,77]. The experimental phases were obtained by identifying the positions of arsenic atoms on the surface of SBC[39], using MR-SAD phases in the PHENIX software suite[78,79]. The structure was reconstructed by AutoBuild[80,81] and manually built in Coot[82]. The resulting structure was refined iteratively with REFMAC[83].

### Crystallization and structure determination of SBC-EpoR.
Five times molar excess of EpoR_pY426 (ASFEpYTILDPS-amide) was incubated with S$^{KKQ}$BC (5 mg ml$^{-1}$). Unbound peptide was removed by a protein concentrator (sartorius Vivaspin) while the mixture was concentrated to 20 mg ml$^{-1}$ concentration. Sodium cacodylate pH7.2 was added to a final concentration of 0.1 M prior to crystallization. Crystallization drops were set up in a ratio of 1:1 protein:precipitant in 18% ethanol, 0.1 M HEPES pH7.5, 0.1 M MgCl$_2$ using hanging drop at 4 °C. Crystals were cryo-protected using 20 % PEG400 prior to flash-cooled.

Diffraction data were collected at 100 K on beamline i24 at Diamond Light Source. Data were recorded to Pilatus3 6M-F detector at 0.97 Å wavelength. Data were indexed, integrated, and reduced using XDS[75] and AIMLESS[76,77]. The phase was obtained by molecular replacement (MR) using Phaser[79] with the coordinates of SOCS2-EloB-EloC (PDB ID: 2C9W) as a search model. The presence of the EpoR_pY426 was observed in the initial electron density map. Model building was conducted manually with Coot[82] and refined with cycles of retrained refinement with REFMAC5[83].

### Crystallization and structure determination of SBC-GHR$_2$.
GHR_pY595 (PVPDpYTSIHIV-amide) and GHR_pY487 (NIDFpYAQVSDI-amide) were mixed with S$^{KKQ}$BC at 1:1:1 stoichiometric ratio with a final concentration of 20 mg ml$^{-1}$ and additional 0.1 M sodium cacodylate pH 7.2. Drops of the complex were mixed 2:1 with 0.005 M cobalt chloride, 0.1 M MES pH6.5 and 1.0 M ammonium sulfate in the sitting-drop vapor diffusion format at 4 °C. 20% MPD was applied to crystal before flash-cooling.

Data collection of the SBC-GHR$_2$ co-crystal was at 100 K on beamline i24 at Diamond Light Source. Images were indexed, intergraded, and reduced using XDS[75] and AIMLESS[76,77]. A molecular replacement solution was obtained by Phaser[79] using SBC-GHR as search model. Refinement was performed using REFMAC5[83] and model building was performed in COOT[82].

### Synthetic details.
All chemicals, unless otherwise stated were commercially available and used without further purification. Solvents were anhydrous and reactions preformed under positive pressure of nitrogen. Flash column chromatography was performed using a Teledyne Isco Combiflash Rf or Rf200i. As pre-packed columns RediSep Rf Normal Phase Disposable Columns were used. NMR spectra were recorded on a Bruker 500 Ultrashield. $^{13}$C spectra were $^1$H decoupled. Chemical shifts ($\delta$) are reported in ppm relative to solvent (CD$_3$OD: $\delta_H = 3.31$ ppm, $\delta_C = 49.0$ ppm) as internal standard. High Resolution Mass Spectra (HRMS) were recorded on a Bruker microTOF. Low resolution MS and analytical HPLC traces (LC-MS) were recorded on an Agilent Technologies 1200 series HPLC connected to an Agilent Technologies 6130 quadrupole LC/MS, connected to an Agilent diode array detector. The column used was a Waters XBridge column (50 mm × 2.1 mm, 3.5 μm particle size) and the compounds were eluted with a gradient of 5−95% acetonitrile/water +0.1% formic acid over 3 min. Preparative HPLC was performed on a Gilson Preparative HPLC System with a Waters X-Bridge C18 column (100 mm × 19 mm; 5 μm particle size) and a gradient of 5% to 95% acetonitrile in water over 10 min, flow 25 ml min$^{-1}$, with 0.1% formic acid in the aqueous phase.

### Cbz-O-bis(dimethylamino)phosphono)-L-tyrosine (1).
O-bis(dimethylamino) phosphono)-L-tyrosine[84] (485 mg, 1.54 mmol) and NaHCO$_3$ (260 mg, 3.1 mmol) were dissolved in the mixture THF/H$_2$O = 1:1 (10 ml) and N-(benzyloxycarbonyloxy)succinimide (383 mg, 1.54 mmol) was added. The reaction mixture was stirred at room temperature overnight. After the addition of 5% NaHSO$_4$ the product was extracted with ethyl acetate, washed with brine, dried over MgSO$_4$, and concentrated by rotary evaporation under reduced pressure. After drying, Cbz-O-bis(dimethylamino)phosphono)-L-tyrosine 1 (620 mg, 89%) was obtained as pale yellow. $^1$H NMR (CD$_3$OD): 2.69 (d, J = 10.1 Hz, 12H), 2.92 (dd, J = 14.0, 9.3 Hz, 1H), 3.18 (dd, J = 14.0, 4.9 Hz, 1H), 4.40 (dd, J = 4.9, 9.3 Hz, 1H), 5.03 (s, 2H), 7.06 (d, J = 8.5 Hz, 2H), 7.21 (d, J = 8.5 Hz, 2H), 7.25–7.36 (m, 5H). $^{31}$P NMR (CD$_3$OD): 18.2. LC-MS (m/z): [M+H]$^+$ calcd. for C$_{21}$H$_{29}$N$_3$O$_6$P, 450.17; found, 450.2.

### Compound 2.
To a mixture of the compound 1 (160 mg, 0.35 mmol), HATU (135 mg, 0.35 mmol), HOAt (48 mg, 0.35 mmol) and DIPEA (150 μl, 1 mmol) in DMF (1 ml), 2 M methylamine solution in THF (0.5 ml) was added under stirring at room temperature. After 2 h, LC-MS analysis showed complete conversion of the

starting material and formation of the desired product. The mixture was diluted with ethyl acetate, washed with 5% NaHSO$_4$, brine, dried over MgSO$_4$, and concentrated by rotary evaporation under reduced pressure. The crude product was dissolved in the mixture ethanol/ethyl acetate = 1:1 (8 ml). Hydrogenation was carried out using H-Cube at 80 °C, Pd/C, 1 atm, at 1 ml min$^{-1}$. The solvent was evaporated under vacuum to afford 2 (108 mg, 92%) which was directly used in the next step without any further purification. $^1$H NMR (CD$_3$OD): 2.66 (s, 3H), 2.71 (d, J = 10.1 Hz, 12H), 2.81 (m, 1H), 2.96 (m, 1H), 3.51 (m, 1H), 7.09 (dd, J = 8.5, 1.0 Hz, 2H), 7.20 (d, J = 8.5 Hz, 2H). $^{31}$P NMR (CD$_3$OD): 18.3. LC-MS (m/z): [M+H]$^+$ calcd. for C$_{14}$H$_{26}$N$_4$O$_3$P, 329.17; found, 329.2.

### Spy molecule 3.
A solution of the compound 2 (80 mg, 0.24 mmol) and DIPEA (85 μl, 0.48 mmol) in DCM (2 ml) was cooled to −78 °C, and trifluoroacetic anhydride (34 μl, 0.24 mmol) was added. The reaction mixture was stirred 1 h at −78 °C. After solvent evaporation the residue was dissolved in acetonitrile (0.5 ml) and 2 M HCl was added (2 ml). The mixture was stirred at room temperature overnight until no presence of the starting materials was detected by LC-MS. The solvents were evaporated and residue was purified by HPLC to afford compound 3 (30 mg, 34%) as a white solid. $^1$H NMR (CD$_3$OD): 2.69 (s, 3H), 2.96 (dd, J = 13.8, 6.4 Hz, 1H), 3.15 (dd, J = 13.8, 6.4 Hz, 1H), 4.57 (dd, J = 8.8, 6.4 Hz, 1H), 7.13 (dd, J = 8.5, 1.1 Hz, 2H), 7.22 (d, J = 8.5 Hz, 2H). $^{13}$C NMR (CD$_3$OD): 26.3, 37.8, 56.5, 117.3 (q, J = 286.7 Hz), 121.3 (d, J = 4.5 Hz), 131.3, 134.1, 151.9 (d, J = 6.8 Hz), 158.7 (q, J = 37.5 Hz), 172.4. $^{31}$P NMR (CD$_3$OD): 3.7. $^{19}$F NMR (CD$_3$OD): −75.6. HRMS (m/z): [M+H]$^+$ calcd. for C$_{12}$H$_{15}$F$_3$N$_2$O$_6$P, 371.0620; found, 371.0599.

### Peptide synthesis.
All peptides were prepared via solid-phase peptide synthesis on 10 mmol scale using standard Fmoc chemistry on Rink amide resin (0.68 mmol g$^{-1}$) on an INTAVIS ResPepSL automated peptide synthesizer. O-(dibenzylphosphono)-N-Fmoc-L-tyrosine was synthesized as described below. The peptides were cleaved with 2.5% triisopropylsilane and 2.5% water in TFA. The crude peptides were isolated from the cleavage mixture by precipitation with cold ether, dissolved in the mixture water/DMF = 1/1 and purified by preparative HPLC under the following conditions: Waters X-Bridge C18 column (100 mm × 19 mm; 5 μm particle size), gradient of 5–95% acetonitrile in water over 10 min, flow 25 mL min$^{-1}$, with 0.1% formic acid in the aqueous phase, UV detection at λobs = 190 and 210 nm. The poorly there soluble peptides were purified according to literature procedure[85]: the impurities were extracted by DCM from the solution of peptides in 20% acetic acid. The purity and identity of the peptides were determined by the analytical LCMS on an Agilent Technologies 1200 series HPLC connected to an Agilent Technologies 6130 quadrupole LC/MS linked to an Agilent diode array detector (raw data shown in Supplementary Fig. 9).

### O-(dibenzylphosphono)-N-Fmoc-L-tyrosine.
To a solution of Fmoc-tyrosine (2 g, 5 mmol) in anhydrous THF (12 ml) N-methylmorpholine (540 μl, 5 mmol) and tert-butyldimethylsilyl chloride (740 mg, 4.9 mmol) were added. After 15 min 4,5-dicyanoimidazole (1.8 g, 15 mmol) and diisopropylphosphoramidite (3.4 ml, 10 mmol) were added and the reaction mixture was stirred at room temperature for 4 h. After cooling to 0 °C 70% tert-butyl hydroperoxide (2 ml, 15 mmol) was introduced. After stirring for 2 h at 0 °C, 10% Na$_2$S$_2$O$_5$ (20 ml) was added and stirring continued for one more hour. The product was extracted with ethyl acetate, washed with a 5% solution of KHSO$_4$, brine, dried over MgSO$_4$, concentrated by rotary evaporation under reduced pressure, and further purified by column chromatography on silica gel using a gradient elution of 0–10% of MeOH in DCM to afford O-(dibenzylphosphono)-N-Fmoc-L-tyrosine (3 g, 90%) as a pale yellow solid. NMR spectra were in agreement with the published data[86].

### Isothermal titration calorimetry.
Experiments were performed with ITC200 instrument (Malvern) in 100 mM HEPES pH 7.5, 50 mM NaCl, 0.5 mM TCEP at 298 K stirring the sample at 750 rpm. The ITC titration consisted of 0.4 μl initial injection (discarded during data analysis) followed by 19 of 2 μl injections at 120 s interval between injections. The GHR_pY595 peptide (PVPDpYTSIHIV-amide, 750 μM), EpoR_pY426 (ASFEpYTILDPS-amide, 750/1500 μM) and GHR_pY487 (NIDFpYAQVSDI-amide, 1500 μM) were directly titrated into SBC (50 μM). Binding data was subtracted from a control titration where peptide was titrated into buffer, and fitted using a one-set-of-site binding model to obtain dissociation constants, binding enthalpy ($\Delta H$), and stoichiometry (N) using MicroCal ITC-ORIGIN Analysis Software 7.0 (Malvern).

### Surface plasmon resonance.
Experiments were performed using Biacore T200 instrument (GH Healthcare) in 20 mM HEPES pH7.5, 150 mM NaCl, 1 mM TCEP, 0.005% Tween20 buffer at 10 °C. Biotinylated wild-type SBC and mutants were immobilized onto a chip surface and injected a series of seven concentrations (0.08, 0.25, 0.7, 2.2, 6.7, 20 and 60 μM) of peptide across the sensor surface for 60 s contact time and 120 sec dissociation time at 30 μl min$^{-1}$ flow rate. Data analysis was carried out using Biacore Evaluation Software (GE Healthcare). All data were double-referenced for reference surface and blank injection. The processed sensograms were fit to a steady-state affinity using a 1:1 binding model for $K_D$ estimation (raw data shown in Supplementary Figs. 10–15).

**$^{19}$F CPMG NMR spectroscopy**. All NMR experiments were conducted using AV-500 MHz Bruker spectrometer equipped with a 5 mm CPQCI 1H/19F/13C/15N/D Z-GRD cryoprobe) at 298 K. Spectra were recorded using 80 scans of a CPMG pulse sequence that attenuates broad resonances. A CPMG delay of 0.133 s was used, to maximize the difference between the signal intensity of spy molecule alone and in the presence of protein (Supplementary Fig. 7). The transmitter frequency was placed close to the resonance of $O_1 = -35451$ Hz ($-75.3$ ppm). Protein was used at 5 µM and spy molecule **3** was used at 100 µM in buffer containing 20 mM HEPES pH 8, 50 mM NaCl, 1 mM DTT, 20% $D_2O$. The GHR related peptides including wild-type, V($-3$)R, V($-3$)Y and alanine scanning peptides were used at 10 µM, whereas the EpoR related peptide including wild-type and alanine scanning were used at 50 µM. All NMR data were processed and analyzed using TopSpin (Bruker).

The dissociation constant of peptides ($K_i$) was calculated by adapting the method described by Wang et al.[47] Briefly, peptides' $K_i$ values were obtained from the equation below:

$$K_i = \frac{([P_0] - [PI] - [PL])([I_0] - [PI])}{[PI]} \quad (1)$$

where $[I_0]$ and $[P_0]$ are the total concentrations of the competitor (peptide inhibitor) and protein, respectively, used in the experiment, while $[PI]$ and $[PL]$ are the free concentrations of protein–peptide and protein–spy complexes, which are unknown.

To determine $[PL]$ the following equation was used:

$$\frac{I_F - I_I}{I_F - I_P} = \frac{[PL]}{[PL_0]} \quad (2)$$

where $I_F$ is the measured integral of the fluorine peak of the spy molecule free in solution; $I_P$ is the integral of the same signal in the presence of protein; $I_I$ is the integral of the same signal in the presence of protein and competing peptide; and $[PL_0]$ is the concentration of protein-spy complex in the absence of competitor, which was calculated using the following equation:

$$[PL_0] = \frac{[P_0] + [L_0] + K_D - \sqrt{([P_0] + [L_0] + K_D)^2 - 4[P_0][L_0]}}{2} \quad (3)$$

where $[L_0]$ is the total concentrations of spy molecule used in the experiment, and $K_D$ is the dissociation constant of the protein-spy complex (determined by ITC). To determine $[PI]$ the following equation was used:

$$[PL] = \frac{[P_0] + [L_0] + K_D - [PI] - \sqrt{([P_0] + [L_0] + K_D - [PI])^2 - 4([P_0] - [PI])[L_0]}}{2} \quad (4)$$

where $[PL]$ was determined as described above (Equation 2).

**$T_2$ relaxation measurement**. Spectra were acquired for each sample of spy alone and spy in the presence of protein (wild-type and SNP mutants) at 298 K, and 80 scans using a CPMG pulse sequence with varying relaxation delays of 0.05, 0.1, 0.2, 0.4, and 0.8 s. Protein was used at 5 µM and spy was used at 100 µM in buffer containing 20 mM HEPES pH 8, 50 mM NaCl, 1 mM DTT, 20 % $D_2O$. The $T_2$ relaxation time was calculated by fitting the data (Supplementary Fig. 7) as a mono exponential decay (GraphPad Prism 6) using the equation below

$$I(t) = I(0)e^{-t/T_2}, \quad (5)$$

where $I(t)$ is the signal intensity or integral at CPMG filter $t$ (in seconds), $I(0)$ is the signal intensity when $t = 0$, and $T_2$ is the time constant of decay.

**Reporting summary**. Further information on research design is available in the Nature Research Reporting Summary linked to this article.

## Data availability

The coordinates and structure factors for SBC in complex with EpoR_pY426 peptide, GHR_pY595 peptide, and GHR2_pY595 peptide have been deposited to the Protein Data Bank (PDB) with accession codes 6I4X, 6I5N, and 6I5J, respectively. The source data underlying Fig. 5b; Tables 2, 3, Supplementary Figs. 1d and 4a are provided as a Source Data file. Other data are available from the corresponding author upon reasonable request.

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

## Acknowledgements

This project has received funding from the European Research Council (ERC) under the European Union's Seventh Framework Program (FP7/2007–2013) as a Starting Grant to A.C. (grant agreement No. ERC-2012-StG-311460 DrugE3CRLs). Biophysics and drug discovery activities at Dundee are supported by Wellcome Trust strategic awards (100476/Z/12/Z and 094090/Z/10/Z, respectively). W.K. was supported by a PhD Scholarship from the School of Life Sciences at the University of Dundee. We thank P. Fyfe for support with

in-house X-ray facilities, the Diamond Light Source for beamtime (BAG proposals MX14980–13 and MX14980–20) and P. Romano and P. Aller for support at beamlines. We thank G. Castro for support with NMR and helpful discussions.

## Author contributions

A.C. conceived the project. W.W.K., S.R., N.M., and A.C. designed experiments. W.W.K., S.R., N.M., and E.B. performed experiments. W.W.K., S.R., N.M., E.B., and A.C. analysed data. W.W.K., S.R., N.M., and A.C. wrote the manuscript.

## Additional information

**Competing interests:** The A.C. laboratory receives sponsored research support from Boehringer Ingelheim and Nurix, Inc. A.C. is a scientific founder, director and shareholder of Amphista Therapeutics, a company that is developing targeted protein degradation therapeutic platforms. The remaining authors declare no competing interests.

