## [Peer Review File · Nature Communications]

Reviewers' comments:

Reviewer #1 (Remarks to the Author):

In this clearly written manuscript, the authors describe structures of SOCS2 bound to phosphopeptides from the Erythropoietin Receptor and Growth Hormone Receptor, which are the first reported structures of SOCS2 bound to phosphopeptide. The BG loop of SOCS2 is also resolved for the first time in one of the structures presented, showing that unlike other SH2 domains, SOCS2 may not use this loop to recognise / lock-in the bound peptide. Collectively these structures begin to unravel how SOCS2 accommodates different targets which will certainly be of interest to both the signalling field and SH2-domain field.

The structure of EpoR peptide bound to SOCS2 reveals a "typical" pY-peptide binding site, and defines the interactions involved. Given the resolution of the data and the low completeness of the data in the outer shell, care should be taken in interpreting the "intricate hydrogen bonding network". Presumably the EF loop moves significantly in the EpoR structure compared to the unbound structure, in order for Ile109 in particular to participate in formation of the hydrophobic pocket? A figure showing this would be helpful.

An intriguing observation is made in co-crystallisation experiments with GHR peptides, in which two copies of peptide are bound to the protein, one in the canonical groove, and one running in an antiparallel conformation making contacts with both the canonical peptide and the protein, forming a beta-sheet. Even when a mixture of two different peptides are used in crystallisation preparations, the same peptide configuration of two copies of the same GHR is seen. This is a very interesting observation, and the authors point out that a similar mode of binding (whereby an SH2 domain binds two peptides in an anti-parallel beta-sheet arrangement) has only been seen once before. Additional figures would be helpful here also – one showing an overlay of EpoR and GHR structures to show similarities / differences in binding in the "normal" site, and an additional panel in Figure 7 comparing the SHP2 structure. A cartoon representation for peptide 7(c) might also be more informative in given the representation in (a) and (b).

My main criticism of this work is that no attempts have been made to validate the GHR peptide structures in solution. If this arrangement can be validated outside the crystal, the significance of the work would be greatly improved.

I'm of the opinion that cell-based experiments are well outside of the scope of this paper, but perhaps there are in vitro experiments that could have been performed. ITC measurements using GHR_pY595 show a stoichiometry of 1:1, implying the affinity of the second site is very weak. If ITC is performed using a 1:1 ratio of SOCS2BC: GHR_pY595, titrating in additional peptide, can any binding be measured? If a peptide is engineered to incorporate two repeating GHR_pY595 motifs with an appropriate length linker, is the affinity of this peptide significantly higher? Or with a synthetic peptide that incorporates both pY595 and pY487 motifs?

The final section of the paper looks at SNPs. L106V and C133Y seem to have little effect on binding. Mutation of R96 and N94 results in significant loss of binding, which would be predicted from the structure. While it is always good to validate assumptions, I'm not sure that the R96Q mutation is "striking" in its loss of affinity. Loss of affinity by mutation of this site has already been shown (Rupp et al. PLoS Genet. 2015). Mutagenesis of protein (rather than peptide) to validate key binding regions might have been more interesting. Are there any disease causing SNPs that might provide circumstantial evidence for the second GHR site (without measurable binding these could not be tested of course)?

Other queries / comments

1. Are there any hints from the EpoR peptide structure that imply it cannot adopt the same two-peptide as GHR? Can the structures tell us if the sequence of the pY487 peptide looks like it would be able to be accommodated in the two-peptide arrangement, either alone or with pY595?
2. SBC-GHR and SBC-GHR2 structures have been indexed differently, is there a reason for this? If not it would be appropriate to reindex to be the same (conventionally $a < b < c$)
3. In table 1, is there a mistake the number of non-hydrogen atoms for ligand in SBC-GHR structure? 94 seems too low. Please check data collection statistics carefully in table one as there are a few (minor) discrepancies between the PDB validation reports and Table 1. An invalid PDB validation report has been submitted for the SBC-EpoR structure. Have the discrepancies between sequence and model as indicated in the PDB report for SBC-GHR been corrected?
4. All binding experiments (ITC, Biacore and NMR) have only been performed once. I would have liked to see duplicate data for all ITC and Biacore experiments, although I acknowledge that for

the majority of measurements two techniques have been used which perhaps circumvents the need for duplicate experiments where these correlate well. Interpretation of GHR pY(+4) data and EpoR pY(+5) data in particular would benefit from duplication (Table 3), as well as GHR_pY487 with WT protein which does not fit the trend and differs from the value cited in the introduction (Figure 6b).

5. The affinities for pY595 and pY487 cited in the introduction are 1.6 μ M and 11.3 μ M, (though measure by the authors in this manuscript as 2.1 μ M and 2.8 μ M respectively, Figure 6b). Depending on the accuracy of these measurements, it may mean that pY595 binding in the canonical site will dominate. If the phosphotyrosine does not contribute to binding in the second site, using PY487 and unphosphorated Y595 peptide might be a way of attempting to obtaining a different two-peptide structure. This is merely a comment; I do not expect another structure to be derived prior to publication.

6. The peptides in table 2 are written with Y instead of pY

Reviewer #2 (Remarks to the Author):

The manuscript "Structural insights into substrate recognition by the SOCS2 E3 ubiquitin ligase" by Kung et al., describes the first structure of SOCS2 bound to a substrate. SOCS2 is one of a family of inhibitors of cytokine signalling (the Suppressors of Cytokine Signalling (SOCS) family). These proteins contain an SH2 domain (to recognise phosphorylated substrates) and a SOCS box domain (to degrade those substrates by interacting with an E3 ubiquitin ligase). SOCS2 specifically suppresses signalling by growth hormone (along with several other cytokines) and SOCS2 $-/-$ mice display gigantism. Although there have been several published structures of SOCS2, including by this group, this is the first time that a structure has been solved with a bound substrate. In this manuscript the Ciuli group solve two such structures, one with a fragment of the growth hormone receptor and the other with a fragment of the EPO receptor. They show, surprisingly, that a single SOCS2 molecule (in the crystal) interacts with two copies of the growth hormone fragment. One is bound in the "canonical" manner, similar to other SOCS proteins and other SH2 domains in general whilst the other is bound in an unusual fashion, in a conformation only previously seen in a peptide bound to one of the SH2 domains of the phosphatase SHP-2. It is notable that shp2 also binds the growth hormone receptor at the same site as SOCS2-do the authors think this may be significant? The authors fully characterise binding both structurally and biophysically and also show that several potentially deleterious SNP's are loss-of-function. In addition to providing novel biological insights this data presented in this manuscript allows the future use of SOCS2 as a PROTAC (once small-molecule binders are identified) and this would yield exciting therapeutic opportunities.

Overall this is well-written and thorough manuscript and will be of interest to those in the signalling field, to those who study SH2 domains in general and to those developing PROTACS, I recommend publication in Nature Communications and have only minor comments/queries.

Introduction:

p2 line 39: It's actually unclear what the purpose of the ESS is in most SOCS proteins, but it is probably not substrate binding (with the exception of SOCS1 and SOCS3).

p2 line 50: cytokines can induce receptor oligomerisation but in many cases induce re-orientation of a pre-existing receptor dimer.

p3 line 66: In regards to the SOCS2 overexpression mouse showing enhanced growth like the KO mouse does. The KO mice are ~40% bigger than WT whereas the OX mice are only ~15% bigger. In any case the reason for this is still unclear but is unlikely to be due to SOCS2-mediated degradation of other SOCS proteins. This was investigated explicitly in Kiu et al., Growth Factors 2009

p5 line 161: Is it possible that cobalt helps stabilise the 2:1 complex? Was SPR etc ever tried with cobalt in the buffer?

p6 line 178: the BG loop connects the B-helix and the G strand not the E and F strands.

Discussion:

page 10, line 285: How distinct are the "distinct hydrophobic cavities"? Is binding dominated by the pY and +3 position (for both peptides) such as seen in other SOCS proteins and in many other SH2 domains? A figure to illustrate this would be beneficial

p11 line 315: SHP2 has two SH2 domains

p13 line 382. Could the authors expand on any potential drawbacks or advantages in using a molecule such as SOCS2 (which is only expressed in response to cytokines) rather than a ubiquitously expressed E3 ligase.

Grammatical/spelling suggestions:

Abstract: "cocystal" should read "co-crystal"

Introduction:

p2 line 39: "associates" should read "associated"

p2 line 51: "docking site" should read "docking sites"

General Comments:

An overlay of the EPOR and GHR peptides (bound to SOCS2) would be beneficial to include as a figure as it allows a qualitative comparison of the two binding sites

Was there any evidence of a 2:1 GHR:SOCS2 stoichiometry in vitro? Zhang et al use NMR to show the formation of such a complex. Given that phosphopeptides with a single ¹⁵N-labelled amino-acid (and all other ¹⁴N) are readily available and inexpensive, one such could be used to monitor formation of a 1:1 and 2:1 complex.

Sincerely

Jeff Babon

Reviewer #3 (Remarks to the Author):

The authors in this study structurally and biophysically characterize the SOCS2 interaction with two different peptide ligands, EpoR and GHR. Direct binding affinities were determined by SPR and complemented with a competitive ¹⁹F NMR displacement assay. Together these data were used to build a model for molecular recognition with the peptide substrates and identify hot spot residues at the interface. This study provides both fundamental insights into the protein-protein interaction and should lead to future ligand discovery efforts to modulate the function of this interaction. The data is compelling although a few additional pieces of experimental data should be added to the supporting information to allow reviewers to more effectively evaluate the data and use the spectral data as a references for future studies. Comments are provided below for the authors to address to help with the clarity and the analyses in their study.

1. For the ¹⁹F NMR displacement assay it was unclear of the duration of each experiment and how many data points were taken. One potential concern is the trifluoroacetamide, which may have appreciable hydrolysis over prolonged experiment times, particularly at pH 8, which would effect the quantitative analysis. Due to the absence of any of the spectral data it is hard to interpret the quality of this data.

At minimum it would be nice to see a representative set of traces of the experimental data both for SPR and ¹⁹F NMR. It would also be useful to include all of the competitive binding isotherms, as additional information can be learned from the plots, such as evaluating hill slopes.

2. The authors noted a 1:2 binding mode in the crystal structure with the GHR peptide, but fit their ITC, SPR and ¹⁹F NMR data based on a 1:1 binding mode. It could simply be that the second binding site is very low affinity, but it would be useful to note how the data changes when a 2:1 binding model is used.

Also related, for the ¹⁹F competition experiment, it is unclear if only one copy of the peptide can compete off the spy molecule, and thus the combined affinity for the second copy is not being determined. The authors could do the experiment in reverse to see if the peptide could be completely competed off by the spy.

3a) Since both SPR and ¹⁹F NMR were only done as single experiments, it was unclear what the error in the measurements were. It would at least be helpful for the experiment to be done in

triplicate for the wildtype peptides.

b) The authors note that the correlation between SPR Kd and F19 NMR Kd match very well. I would suggest adding the R2 value to the main text to be more quantitative. Although, I agree this correlation looks very good!

4. The authors should include the characterization data for their peptides. E.g HPLC purity traces, and MS confirmatory data. For the small molecules, the spectral data is tabulated but the individual spectra are absent.

5. Figure S5 shows stacked 1H spectra of the proteins suggesting minimal perturbation of structure. This is a pretty low resolution experiment, and the amide region is barely visible. I would suggest blowing up that region of the spectra, and potentially overlaying the spectra. Without that, it is difficult to critically interpret the data.

Minor: In the main text, I would suggest referring to the NMR Kd as a Ki (as in the experimental) as its derived from a competition experiment.

Response to Reviewers

Reviewer #1 (Remarks to the Author):

In this clearly written manuscript, the authors describe structures of SOCS2 bound to phosphopeptides from the Erythropoietin Receptor and Growth Hormone Receptor, which are the first reported structures of SOCS2 bound to phosphopeptide. The BG loop of SOCS2 is also resolved for the first time in one of the structures presented, showing that unlike other SH2 domains, SOCS2 may not use this loop to recognise / lock-in the bound peptide. Collectively these structures begin to unravel how SOCS2 accommodates different targets which will certainly be of interest to both the signalling field and SH2-domain field.

The structure of EpoR peptide bound to SOCS2 reveals a “typical” pY-peptide binding site, and defines the interactions involved. Given the resolution of the data and the low completeness of the data in the outer shell, care should be taken in interpreting the “intricate hydrogen bonding network”. Presumably the EF loop moves significantly in the EpoR structure compared to the unbound structure, in order for Ile109 in particular to participate in formation of the hydrophobic pocket? A figure showing this would be helpful.

An intriguing observation is made in co-crystallisation experiments with GHR peptides, in which two copies of peptide are bound to the protein, one in the canonical groove, and one running in an antiparallel conformation making contacts with both the canonical peptide and the protein, forming a beta-sheet. Even when a mixture of two different peptides are used in crystallisation preparations, the same peptide configuration of two copies of the same GHR is seen. This is a very interesting observation, and the authors point out that a similar mode of binding (whereby an SH2 domain binds two peptides in an anti-parallel beta-sheet arrangement) has only been seen once before. Additional figures would be helpful here also – one showing an overlay of EpoR and GHR structures to show similarities / differences in binding in the “normal” site, and an additional panel in Figure 7 comparing the SHP2 structure. A cartoon representation for peptide 7(c) might also be more informative in

given the representation in (a) and (b).

My main criticism of this work is that no attempts have been made to validate the GHR peptide structures in solution. If this arrangement can be validated outside the crystal, the significance of the work would be greatly improved.

I’m of the opinion that cell-based experiments are well outside of the scope of this paper, but perhaps there are in vitro experiments that could have been performed. ITC measurements using GHR_pY595 show a stoichiometry of 1:1, implying the affinity of the second site is very weak. If ITC is performed using a 1:1 ratio of SOCS2BC: GHR_pY595, titrating in additional peptide, can any binding be measured? If a peptide is engineered to incorporate two repeating GHR_pY595 motifs with an appropriate length linker, is the affinity of this peptide significantly higher? Or with a synthetic peptide that incorporates both pY595 and PY487 motifs?

The final section of the paper looks at SNPs. L106V and C133Y seem to have little effect on binding. Mutation of R96 and N94 results in significant loss of binding, which would be predicted from the structure. While it is always good to validate assumptions, I’m not sure that the R96Q mutation is “striking” in its loss of affinity. Loss of affinity by mutation of this site has already been shown (Rupp et al. PLoS Genet. 2015). Mutagenesis of protein (rather than peptide) to validate key binding regions might have been more interesting. Are there any disease causing SNPs that might provide circumstantial evidence for the second GHR site (without measurable binding these could not be tested of course)?

Other queries / comments

1. Are there any hints from the EpoR peptide structure that imply it cannot adopt the same two-peptide as GHR? Can the structures tell us if the sequence of the pY487 peptide looks like it would be able to be accommodated in the two-peptide arrangement, either alone or with pY595?
2. SBC-GHR and SBC-GHR2 structures have been indexed differently, is there a reason for this? If

- not it would be appropriate to reindex to be the same (conventionally $a < b < c$)
- In table 1, is there a mistake the number of non-hydrogen atoms for ligand in SBC-GHR structure? 94 is seems too low. Please check data collection statistics carefully in table one as there are a few (minor) discrepancies between the PDB validation reports and Table1. An invalid PDB validation report has been submitted for the SBC-EpoR structure. Have the discrepancies between sequence and model as indicated in the PDB report for SBC-GHR been corrected?
 - All binding experiments (ITC, Biacore and NMR) have only been performed once. I would have liked to see duplicate data for all ITC and Biacore experiments, although I acknowledge that for the majority of measurements two techniques have been used which perhaps circumvents the need for duplicate experiments where these correlate well. Interpretation of GHR pY(+4) data and EpoR pY(+5) data in particular would benefit from duplication (Table 3), as well as GHR_pY487 with WT protein which does not fit the trend and differs from the value cited in the introduction (Figure 6b).
 - The affinities for pY595 and pY487 cited in the introduction are $1.6\mu\text{M}$ and $11.3\mu\text{M}$, (though measure by the authors in this manuscript as $2.1\mu\text{M}$ and $2.8\mu\text{M}$ respectively, Figure 6b). Depending on the accuracy of these measurements, it may mean that pY595 binding in the canonical site will dominate. If the phosphotyrosine does not contribute to binding in the second site, using PY487 and unphosphorated Y595 peptide might be a way of attempting to obtaining a different two-peptide structure. This is merely a comment; I do not expect another structure to be derived prior to publication.
 - The peptides in table 2 are written with Y instead of pY

Response to reviewer 1:

1.1 Additional figures would be helpful here also – one showing an overlay of EpoR and GHR structures to show similarities / differences in binding in the “normal” site, and an additional panel in Figure 7 comparing the SHP2 structure. A cartoon representation for peptide 7(c) might also be more informative in given the representation in (a) and (b).

We appreciate the suggestion and we concur. To address this point, we have included a new Figure 6a-b:

and added the following text:

“Residues at pY(-1), pY(+1) and pY (+3) positions of the GHR_pY595 and EpoR_pY426 have similar properties; whereas residues at pY(-3), pY(+2) and pY(+5) position differ in properties and sizes (Figure 6a).”

We also include a new Figure 7c-e

and added to the text:

“ A comparison of SOCS2-GHR structure with SHP2 structure (PDB-3TKZ) reveals a conserved SH2 domain with two copies of the substrate peptides binding SOCS2 and SHP2. In both the structures, the BG loop folds up as a hairpin and forms a triple triple-stranded β sheet interaction with two bound peptides, which we will further elaborate later (Figure 7d). Although the canonical phosphotyrosine binding site is conserved for both SOCS2 and SHP2, the positioning of the non-canonical phosphotyrosine varies significantly. Comparison of distance between CA atoms of phosphotyrosines of the two peptides in the SBC-GHR and SHP-2 substrates: It is 23Å and 13Å in GHR and SHP2-substrate respectively. Longer distance between the canonical and non-canonical pY enable antiparallel beta strand interactions between 4 amino acids GHR peptides unlike antiparallel interactions between two amino acids in SHP-2. Furthermore, a longer EF loop in SOCS2 pushes the two GHR peptide, away from itself towards the open BG loop, enabling β sheet formation.”

1.2 My main criticism of this work is that no attempts have been made to validate the GHR peptide structures in solution. If this arrangement can be validated outside the crystal, the significance of the work would be greatly improved.

We appreciate the comment and agree that validation of co-crystal structures with binding data in solution is important. We include substantial data in solution by NMR, ITC and SPR including mutagenesis that we believe provides robust validation for our crystal structure main binding mode. As discussed and noted (see also below response to point 3.2 from reviewer 3) our biophysical data in solution supports the 1:1 mode also for GHR peptide, and suggests that the second binding mode is of low affinity. Probing this second binding mode also in solution will be the subject of future investigations.

1.3 I'm of the opinion that cell-based experiments are well outside of the scope of this paper, but perhaps there are in vitro experiments that could have been performed. ITC measurements using GHR_pY595 show a stoichiometry of 1:1, implying the affinity of the second site is very weak. If ITC is performed using a 1:1 ratio of SOCS2BC: GHR_pY595, titrating in additional peptide, can any binding be measured? If a peptide is engineered to incorporate two repeating GHR_pY595 motifs with an appropriate length linker, is the affinity of this peptide significantly higher? Or with a synthetic peptide that incorporates both pY595 and PY487 motifs?

These are good points. We have performed several replicates of ITC with GHR_pY595, also in varying conditions, and have in all cases observed 1:1 stoichiometry (four replicates, see Supp Info Figure S1). Our data therefore is consistent with the second binding mode being of low affinity/heat of binding.

The suggestion of merging the two sequences in a single peptide is a valuable suggestion, and a direction that we would endeavor to pursue in future work.

1.4 The final section of the paper looks at SNPs. L106V and C133Y seem to have little effect on binding. Mutation of R96 and N94 results in significant loss of binding, which would be predicted from the structure. While it is always good to validate assumptions, I'm not sure that the R96Q mutation is "striking" in its loss of affinity. Loss of affinity by mutation of this site has already been shown (Rupp et al. PLoS Genet. 2015).

Thank you for this comment. We have amended the text: "*Mutations at R96 site abrogate SOCS2 binding was also reported by Rupp et al. with R96C*".

1.5 Mutagenesis of protein (rather than peptide) to validate key binding regions might have been more interesting. Are there any disease causing SNPs that might provide circumstantial evidence for the second GHR site (without measurable binding these could not be tested of course)?

The SNPs on SOCS2 and GHR reported from COSMIC database are highlighted in red. We did not find any SNP around the second GHR site.

1.6 Other queries / comments: Are there any hints from the EpoR peptide structure that imply it cannot adopt the same two-peptide as GHR? Can the structures tell us if the sequence of the pY487 peptide looks like it would be able to be accommodated in the two-peptide arrangement, either alone or with pY595?

All very valuable questions, thank you. We have added to the text:
“Binding of the non-canonical GHR peptide is driven by backbone to backbone hydrogen bonding interaction with the canonical GHR peptide and BG loop (Figure 6d). The 1:1 binding mode for EpoR could be justified by the presence of Pro(+5) that acts as a strand breaker and prevents backbone to backbone interaction with the second peptide. Indeed, from the alanine scan, we observe that when Pro(+5) is replaced by Alanine, the affinity actually improves - implying the possibility of secondary peptide binding.”

And added new Figure 6d:

Legend added: *“(d) The binding of the non-canonical binding GHR (cyan) relies on backbone to backbone interaction and hydrophobic interaction from pY(+3) and pY(+5) of the GHR_pY595 to the SH2 domain hydrophobic core (pink). The binding mode of EpoR (orange) makes it unfavourable to establish these interactions. ”*

The following are sequences of GHR_pY595 and pY487.

	-4	-3	-2	-1	0	+1	+2	+3	+4	+5	+6
GHR_pY595	P	V	P	D	pY	T	S	I	H	I	V
GHR_pY487	N	I	D	F	pY	A	Q	V	S	D	I
EpoR	A	S	F	E	pY	T	I	L	D	P	S

We do not feel the current data provides sufficient evidence to make statements with regard to the question on the binding mode (1:1 or 2:1) for GHR_pY487. The residue at pY(+2) in GHR_pY487 is bulkier, and may provoke a rearrangement of EF loop. At the pY (+3), both residues remain hydrophobic to catch the interaction at the hydrophobic core of SH2 domain. However, absence of Proline at the pY(+5) position does not eliminate the possibility of formation of beta sheet formation with secondary GHR_pY487 but would require significant conformation change in EF loop and BG loop.

1.7 SBC-GHR and SBC-GHR2 structures have been indexed differently, is there a reason for this? If not it would be appropriate to reindex to be the same (conventionally a<b<c)

The software failed to index to the same spacegroup. The two structures pack differently in the crystal and makes different contacts.

Symmetry mates of SBC-GHR₂ (brown) and SBC-GHR (green).

1.8 In table 1, is there a mistake the number of non-hydrogen atoms for ligand in SBC-GHR structure? 94 is seems too low. Please check data collection statistics carefully in table one as there are a few (minor) discrepancies between the PDB validation reports and Table1. An invalid PDB validation report has been submitted for the SBC-EpoR structure. Have the discrepancies between sequence and model as indicated in the PDB report for SBC-GHR been corrected?

The table has now been corrected. Final PDB validation reports for the three structures are now provided. The discrepancy between the sequence and the model is because of

the surface entropy reduction mutations in SOCS2 -K115A/K117A/Q118A (highlighted in Methods) that were used to obtain complex structures.

1.9 All binding experiments (ITC, Biacore and NMR) have only been performed once. I would have liked to see duplicate data for all ITC and Biacore experiments, although I acknowledge that for the majority of measurements two techniques have been used which perhaps circumvents the need for duplicate experiments where these correlate well. Interpretation of GHR pY(+4) data and EpoR pY(+5) data in particular would benefit from duplication (Table 3), as well as GHR_pY487 with WT protein which does not fit the trend and differs from the value cited in the introduction (Figure 6b).

ITC and SPR experiments are now reported in repeats. As stated in the legends of Figure 5(b), Table 2 and Table 3 the K_d is now reported as mean \pm s.e.m. from four independent experiments. Figure S1(d) and Figure S4(a) now reports the data as mean \pm s.e.m, with number of independent experiments as a column in the tables.

1.10 The affinities for pY595 and pY487 cited in the introduction are 1.6 μ M and 11.3 μ M, (though measure by the authors in this manuscript as 2.1 μ M and 2.8 μ M respectively, Figure 6b). Depending on the accuracy of these measurements, it may mean that pY595 binding in the canonical site will dominate. If the phosphotyrosine does not contribute to binding in the second site, using PY487 and unphosphorated Y595 peptide might be a way of attempting to obtaining a different two-peptide structure. This is merely a comment; I do not expect another structure to be derived prior to publication.

Data has been added to supplementary Figure S1

We determined the K_d of GHR_pY487 using ITC ($n = 3$) and SPR ($n = 3$) obtaining a K_d of 2.3 and 2.8 μ M, respectively. GHR_pY595 demonstrates higher affinity to SOCS2 ($K_d = 1.1 \mu$ M) suggesting this peptide will be more dominant in binding at the canonical site.

1.11 The peptides in table 2 are written with Y instead of pY

Corrected.

Reviewer #2 (Remarks to the Author):

The manuscript "Structural insights into substrate recognition by the SOCS2 E3 ubiquitin ligase" by Kung et al., describes the first structure of SOCS2 bound to a substrate. SOCS2 is one of a family of inhibitors of cytokine signalling (the Suppressors of Cytokine Signalling (SOCS) family). These proteins contain an SH2 domain (to recognise phosphorylated substrates) and a SOCS box domain (to degrade those substrates by interacting with an E3 ubiquitin ligase). SOCS2 specifically suppresses signalling by growth hormone (along with several other cytokines) and SOCS2 $-/-$ mice display gigantism. Although there have been several published structures of SOCS2, including by this group, this is the first time that a structure has been solved with a bound substrate. In this manuscript the Ciuli group solve two such structures, one with a fragment of the growth hormone receptor and the other with a fragment of the EPO receptor. They show, surprisingly, that a single SOCS2 molecule (in the crystal) interacts with two copies of the growth hormone fragment. One is bound in the "canonical" manner, similar to other SOCS proteins and other SH2 domains in general whilst the other is bound in an unusual fashion, in a conformation only previously seen in a peptide bound to one of the SH2 domains of the phosphatase SHP-2. It is notable that shp2 also binds the growth hormone receptor at the same site as SOCS2-do the authors think this may be significant? The authors fully characterise binding both structurally and biophysically and also show that several potentially deleterious SNP's are loss-of-function. In addition to providing novel biological insights this data presented in this manuscript allows the future use of SOCS2 as a PROTAC (once small-molecule binders are identified) and this would yield exciting therapeutic opportunities.

Overall this is well-written and thorough manuscript and will be of interest to those in the signalling field, to those who study SH2 domains in general and to those developing PROTACS, I recommend publication in Nature Communications and have only minor comments/queries.

Introduction:

p2 line 39: It's actually unclear what the purpose of the ESS is in most SOCS proteins, but it is probably not substrate binding (with the exception of SOCS1 and SOCS3).

p2 line 50: cytokines can induce receptor oligomerisation but in many cases induce re-orientation of a pre-existing receptor dimer.

p3 line 66: In regards to the SOCS2 overexpression mouse showing enhanced growth like the KO mouse does. The KO mice are ~40% bigger than WT whereas the OX mice are only ~15% bigger. In any case the reason for this is still unclear but is unlikely to be due to SOCS2-mediated degradation of other SOCS proteins. This was investigated explicitly in Kiu et al., Growth Factors 2009

p5 line 161: Is it possible that cobalt helps stabilise the 2:1 complex? Was SPR etc ever tried with cobalt in the buffer?

p6 line 178: the BG loop connects the B-helix and the G strand not the E and F strands.

Discussion:

page 10, line 285: How distinct are the "distinct hydrophobic cavities"? Is binding dominated by the pY and +3 position (for both peptides) such as seen in other SOCS proteins and in many other SH2 domains? A figure to illustrate this would be beneficial

p11 line 315: SHP2 has two SH2 domains

p13 line 382. Could the authors expand on any potential drawbacks or advantages in using a molecule such as SOCS2 (which is only expressed in response to cytokines) rather than a ubiquitously expressed E3 ligase.

Grammatical/spelling suggestions:

Abstract: "cocystal" should read "co-crystal"

Introduction:

p2 line 39: "associates" should read "associated"

p2 line 51: "docking site" should read "docking sites"

General Comments:

An overlay of the EPOR and GHR peptides (bound to SOCS2) would be beneficial to include as a figure as it allows a qualitative comparison of the two binding sites

Was there any evidence of a 2:1 GHR:SOCS2 stoichiometry in vitro? Zhang et al use NMR to show the formation of such a complex. Given that phosphopeptides with a single ¹⁵N-labelled amino-acid (and all other ¹⁴N) are readily available and inexpensive, one such could be used to monitor formation of a 1:1 and 2:1 complex.

Sincerely

Jeff Babon

Reviewer 2:

2.1 It is notable that shp2 also binds the growth hormone receptor at the same site as SOCS2-do the authors think this may be significant?

Despite SOCS2 and SHP2 both recognize GHR at the same sites (GHR_pY595 and GHR_pY426) and bound with two peptides, we don't think this is very significant for now, because

1. The two peptides bound in the SHP2 were reported as a class IV peptide. From the protein blast result, this peptide may be adaptor complexes medium subunit family from Plasmodium ovale wallikeri, which is not relevant to GHR.
2. SHP2 binds the GHR at the same sites as SOCS2 but their affinity cannot be found by authors' best effort. Therefore, whether SHP2 also recognizes two GHR is unknown.

Peptide sequences

GHR_pY595: PVPDpYTSIHIV

Class IV peptide: RVIpYFVPLNR

2.2 Introduction:

p2 line 39: It's actually unclear what the purpose of the ESS is in most SOCS proteins, but it is probably not substrate binding (with the exception of SOCS1 and SOCS3).

We have added to the text:

"extended SH2 subdomain (ESS) that functions as a bridge at the interface between the SH2 domain and the SOCS box enabling ubiquitination of captured substrate³⁻⁵"

2.3 p2 line 50: cytokines can induce receptor oligomerisation but in many cases induce re-orientation of a pre-existing receptor dimer.

We have added the following text:

"Upon cytokine binding, the oligomerized receptors activate the JAK family kinases that phosphorylate specific tyrosine residues on the receptor, including the docking sites for the STAT proteins."

2.4 p3 line 66: In regards to the SOCS2 overexpression mouse showing enhanced growth like the KO mouse does. The KO mice are ~40% bigger than WT whereas the OX mice are only ~15% bigger. In any case the reason for this is still unclear but is unlikely to be due to SOCS2-mediated degradation of other SOCS proteins. This was investigated explicitly in Kiu et al., Growth Factors 2009

Amended text:

“SOCS2 has been shown as the primary suppressor of growth hormone (GH) pathway where a gigantism phenotype was observed in a SOCS2^{-/-} mice²⁶. Paradoxically, the SOCS2 overexpressed transgenic mice also led to the same phenotype²⁷.”

2.5 p5 line 161: Is it possible that cobalt helps stabilise the 2:1 complex? Was SPR etc ever tried with cobalt in the buffer?

Thank you for this suggestion. The cobalt only makes two interactions in the structure, one to the His149 of protein, the other one to the His(+4) of the non-canonical GHR peptide. We performed an ITC experiment with 0.75 mM cobalt chloride in the syringe along with GHR_595 for comparison. From the data, the stoichiometry detected by ITC is still 1.

SBC v/s GHR_pY595	No. of trials	K_d (μM)	ΔG ($\text{kcal} \times \text{mol}^{-1}$)	ΔH ($\text{kcal} \times \text{mol}^{-1}$)	$T\Delta S$ ($\text{kcal} \times \text{mol}^{-1}$)	Stoichiometry (N)
No cobalt	4	1.11 ± 0.15	-8.12 ± 0.08	-3.48 ± 0.07	4.63 ± 0.11	1.03 ± 0.03
+ 0.75 mM cobalt	1	1.02 ± 0.26	-8.17 ± 0.15	-3.62 ± 0.10	4.56 ± 0.18	0.98 ± 0.02

2.6 p6 line 178: the BG loop connects the B-helix and the G strand not the E and F strands.

Amended text:

“The BG loop connects the αB and βG strands of an SH2 domain (Figure 3a)”

2.7 Discussion:

page 10, line 285: How distinct are the “distinct hydrophobic cavities”? Is binding

dominated by the pY and +3 position (for both peptides) such as seen in other SOCS proteins and in many other SH2 domains? A figure to illustrate this would be beneficial

Added to the text:

“Residues at pY(-1), pY(+1) and pY (+3) positions of the GHR_pY595 and EpoR_pY426 peptide contains similar properties; whereas residues at pY(-3), pY(+2) and pY(+5) position are different in properties and sizes (Figure 6a). In particular, the Val(-3) of GHR_pY595 and Pro(+5) of EpoR_pY426 peptide catch different hydrophobic interactions resulting in exclusive binding modes in SOCS2 comparing to the substrate peptides of SOCS3 and SOCS6^{4,21} (Figure 6b, 6c).”

New Figure 6c:

“ (c) Overlay of the substrate peptide of SOCS proteins. The GHR_pY595 (yellow) and EpoR (orange) peptides catch distinct hydrophobic cavities (pink) on SOCS2. These interactions differ SOCS2 substrate binding modes from other substrates of SOCS proteins. The SOCS3 substrate peptide gp130 is in cyan (PDB ID: 2HMH) and SOCS6 substrate peptide c-Kit is in green (PDB ID: 2VIF).”

2.8 p11 line 315: SHP2 has two SH2 domains

Amended text:

“a tyrosine phosphatase SHP-2, which contains SH2 domains, is observed a comparable structure”

2.9 p13 line 382. Could the authors expand on any potential drawbacks or advantages in using a molecule such as SOCS2 (which is only expressed in response to cytokines) rather than a ubiquitously expressed E3 ligase.

We appreciate the comment and have included the following text to the manuscript:

“The usage of SOCS2 binders are limited to SOCS2 expressing cells, however, this provides an additional layer of tissue specific degradation of target proteins. PROTACs designed using SOCS2 binders as recruiters would be ideal for degrading targets specifically in cells affected with leukemia and gastrointestinal sarcoma which have been recorded to have upregulated SOCS2 expression⁸⁷. Low levels of SOCS2 in other normal tissues would help minimize toxicity. In addition, a pan-SOCS recruiting PROTACs would help diversifying the target tissue range.”

2.10 Grammatical/spelling suggestions: Abstract: “cocystal” should read “co-crystal”
Corrected.

2.11 Introduction: p2 line 39: “associates” should read “associated”
Corrected.

2.12 p2 line 51: “docking site” should read “docking sites”
Corrected.

2.13 General Comments:

An overlay of the EPOR and GHR peptides (bound to SOCS2) would be beneficial to include as a figure as it allows a qualitative comparison of the two binding sites

New Figures 6a-b

2.14 Was there any evidence of a 2:1 GHR:SOCS2 stoichiometry in vitro? Zhang et al use NMR to show the formation of such a complex. Given that phosphopeptides with a single ^{15}N -labelled amino-acid (and all other ^{14}N) are readily available and inexpensive, one such could be used to monitor formation of a 1:1 and 2:1 complex.

This is a good suggestion, and a direction that we would endeavor to pursue in future work

Reviewer #3 (Remarks to the Author):

The authors in this study structurally and biophysically characterize the SOCS2 interaction with two different peptide ligands, EpoR and GHR. Direct binding affinities were determined by SPR and complemented with a competitive ¹⁹F NMR displacement assay. Together these data were used to build a model for molecular recognition with the peptide substrates and identify hot spot residues at the interface. This study provides both fundamental insights into the protein-protein interaction and should lead to future ligand discovery efforts to modulate the function of this interaction. The data is compelling although a few additional pieces of experimental data should be added to the supporting information to allow reviewers to more effectively evaluate the data and use the spectral data as a references for future studies. Comments are provided below for the authors to address to help with the clarity and the analyses in their study.

1. For the ¹⁹F NMR displacement assay it was unclear of the duration of each experiment and how many data points were taken. One potential concern is the trifluoroacetamide, which may have appreciable hydrolysis over prolonged experiment times, particularly at pH 8, which would effect the quantitative analysis. Due to the absence of any of the spectral data it is hard to interpret the quality of this data.

At minimum it would be nice to see a representative set of traces of the experimental data both for SPR and ¹⁹F NMR. It would also be useful to include all of the competitive binding isotherms, as additional information can be learned from the plots, such as evaluating hill slopes.

2. The authors noted a 1:2 binding mode in the crystal structure with the GHR peptide, but fit their ITC, SPR and ¹⁹F NMR data based on a 1:1 binding mode. It could simply be that the second binding site is very low affinity, but it would be useful to note how the data changes when a 2:1 binding model is used.

Also related, for the ¹⁹F competition experiment, it is unclear if only one copy of the peptide can compete off the spy molecule, and thus the combined affinity for the second copy is not being determined. The authors could do the experiment in reverse to see if the peptide could be completely competed off by the spy.

3a) Since both SPR and ¹⁹F NMR were only done as single experiments, it was unclear what the error in the measurements were. It would at least be helpful for the experiment to be done in triplicate for the wildtype peptides.

b) The authors note that the correlation between SPR K_d and ¹⁹F NMR K_d match very well. I would suggest adding the R² value to the main text to be more quantitative. Although, I agree this correlation looks very good!

4. The authors should include the characterization data for their peptides. E.g HPLC purity traces, and MS confirmatory data. For the small molecules, the spectral data is tabulated but the individual spectra are absent.

5. Figure S5 shows stacked ¹H spectra of the proteins suggesting minimal perturbation of structure. This is a pretty low resolution experiment, and the amide region is barely visible. I would suggest blowing up that region of the spectra, and potentially overlaying the spectra. Without that, it is difficult to critically interpret the data.

Minor: In the main text, I would suggest referring to the NMR K_d as a K_i (as in the experimental) as its derived from a competition experiment.

Reviewer 3:

3.1 For the ^{19}F NMR displacement assay it was unclear of the duration of each experiment and how many data points were taken. One potential concern is the trifluoroacetamide, which may have appreciable hydrolysis over prolonged experiment times, particularly at pH 8, which would effect the quantitative analysis. Due to the absence of any of the spectral data it is hard to interpret the quality of this data.

At minimum it would be nice to see a representative set of traces of the experimental data both for SPR and ^{19}F NMR. It would also be useful to include all of the competitive binding isotherms, as additional information can be learned from the plots, such as evaluating hill slopes.

Methods updated:

"The ^{19}F NMR displacement assay takes one data point at a delay of 0.133 seconds for 80 scans. For the T_2 measurement, 5 data points were taken for each sample using a CPMG pulse sequence with a relaxation delay of 0.05, 0.1, 0.2, 0.4 and 0.8 s for 80 scans"

Trifluoroacetamide bonds are very stable to hydrolysis at room temperature. We performed a stability test of our spy molecule in the exactly same condition as we used for our experiments, plus a 2,2,2-trifluoroethanol as an internal reference (see picture). After 24 h, the ratio of spy/trifluoroethanol is the same as at 0h and there is no presence of other signals suggesting the stability of compounds.

Stability test. SPY + trifluoroethanol, as an internal reference, in NMR buffer (20mM HEPES, pH8, 50mM NaCl and 1mM DTT).

The above new data has been added to supplementary Figure S4 as panel (d)

In addition, the individual SPR sensograms are added in Figures S9-S14

3.2 The authors noted a 1:2 binding mode in the crystal structure with the GHR peptide, but fit their ITC, SPR and F19 NMR data based on a 1:1 binding mode. It could simply be that the second binding site is very low affinity, but it would be useful to note how the data changes when a 2:1 binding model is used.

Also related, for the 19F competition experiment, it is unclear if only one copy of the peptide can compete off the spy molecule, and thus the combined affinity for the second copy is not being determined. The authors could do the experiment in reverse to see if the peptide could be completely competed off by the spy.

We tried fitting the ITC data for SBC-GHR_pY595 using a sequential binding model. However the more complex model does not significantly improve the fitting – see below.

Binding Model	K_{d1} (μM)	ΔH_1 ($\text{kcal} \times \text{mol}^{-1}$)	$T\Delta S_1$ ($\text{kcal} \times \text{mol}^{-1}$)	K_{d2} (μM)	ΔH_2 ($\text{kcal} \times \text{mol}^{-1}$)	$T\Delta S_2$ ($\text{kcal} \times \text{mol}^{-1}$)
Sequential	1.19 ± 0.30	-3.61 ± 0.09	4.47 ± 0.17	847 ± 589	-0.83 ± 1.33	3.34 ± 1.39
1:1	1.11 ± 0.15	-3.48 ± 0.07	4.63 ± 0.08	n/a	n/a	n/a

With regards to the second question: as our data is consistent with the first peptide-binding site being of higher affinity, we believe that both the spy molecule and the displacing peptide bind preferentially to the canonical pY binding site. Therefore, it is the affinity at this site only that gets measured in our competitive NMR experiments.

3.3 a) Since both SPR and 19F NMR were only done as single experiments, it was unclear what the error in the measurements were. It would at least be helpful for the experiment to be done in triplicate for the wildtype peptides.

b) The authors note that the correlation between SPR K_d and F19 NMR K_d match very well. I would suggest adding the R² value to the main text to be more quantitative. Although, I agree this correlation looks very good!

All SPR, ITC and NMR experiments are n ≥ 3. As stated in the legends of Figure 5(b), Table 2 and Table 3 the K_d is now reported as mean ± s.e.m. from four independent experiments. Figure S1(d) and Figure S4(a) now reports the data as mean ± s.e.m, with number of independent experiments as a column in the tables.

a) The two assays were found to be robust and reliable where the measured K_d and K_i values at R² of 0.74 correlation (Figure S5).

3.4 The authors should include the characterization data for their peptides. E.g HPLC purity traces, and MS confirmatory data. For the small molecules, the spectral data is tabulated but the individual spectra are absent.

Added to the supplementary as:

Figure S4(b,c) - NMR spectra (1H and 13C) of spy molecule

Figure S7-LCMS reports (HPLC purity and MS confirmation data)

3.5 Figure S5 shows stacked 1H spectra of the proteins suggesting minimal perturbation of structure. This is a pretty low resolution experiment, and the amide region is barely visible. I would suggest blowing up that region of the spectra, and potentially overlaying the spectra. Without that, it is difficult to critically interpret the data.

Added two zoom-in spectra (10 to 6 ppm region and 1.5 to -0.5 ppm region) to Figure S6.

3.6 Minor: In the main text, I would suggest referring to the NMR K_d as a K_i (as in the experimental) as its derived from a competition experiment.

Corrected

REVIEWERS' COMMENTS:

Reviewer #1 (Remarks to the Author):

The majority of my concerns have been addressed, I believe the article is suitable for publication

Reviewer #3 (Remarks to the Author):

I am satisfied with the author's addition to the manuscript.

The inclusion of the replicate data with error shows a remarkable level of precision.

As a minor area to improve the clarity, on line 270 the authors note "In addition to SPR, the SNP mutants were characterized using ^{19}F NMR by monitoring the transverse relaxation rate (R_2). L106V and C133Y exhibited similar rate, 10 s⁻¹ and 14 s⁻¹, comparable to wild type at 11 s⁻¹ .

I would note that include that its the relaxation rate from the fluorinated spy molecule, and the not the protein. Adding the value of the relaxation rate for just the small molecule from Figure S8 would also help support the argument.

REVIEWERS' COMMENTS:

Reviewer #1 (Remarks to the Author):

The majority of my concerns have been addressed, I believe the article is suitable for publication

Reviewer #3 (Remarks to the Author):

I am satisfied with the author's addition to the manuscript.

The inclusion of the replicate data with error shows a remarkable level of precision.

As a minor area to improve the clarity, on line 270 the authors note "In addition to SPR, the SNP mutants were characterized using ^{19}F NMR by monitoring the transverse relaxation rate (R_2). L106V and C133Y exhibited similar rate, 10 s^{-1} and 14 s^{-1} , comparable to wild type at 11 s^{-1} .

I would note that include that its the relaxation rate from the fluorinated spy molecule, and the not the protein. Adding the value of the relaxation rate for just the small molecule from Figure S8 would also help support the argument.

Thank you for highlighting this issue. We have now revised the manuscript text on pg. 9-10 to aid clarity on this point.